# Telomere length in offspring is determined by mitochondrial-nuclear communication at fertilization

Yasmyn E. Winstanley[1], Ryan D. Rose[1,2], Alexander P. Sobinoff[3], Linda L. Wu[1], Deepak Adhikari[4], Qing-Hua Zhang[4], Jadon K. Wells[3], Lee H. Wong[5], Hazel H. Szeto[6], Sandra G. Piltz[1,7], Paul Q. Thomas[1,7], Mark A. Febbraio[8], John Carroll[4], Hilda A. Pickett[3], Darryl L. Russell[1] & Rebecca L. Robker[1,4] ✉

The initial setting of telomere length during early life in each individual has a major influence on lifetime risk of aging-associated diseases; however there is limited knowledge of biological signals that regulate inheritance of telomere length, and whether it is modifiable is not known. We now show that when mitochondrial activity is disrupted in mouse zygotes, via exposure to 20% $O_2$ or rotenone, telomere elongation between the 8-cell and blastocyst stage is impaired, with shorter telomeres apparent in the pluripotent Inner Cell Mass (ICM) and persisting after organogenesis. Identical defects of elevated mtROS in zygotes followed by impaired telomere elongation, occurred with maternal obesity or advanced age. We further demonstrate that telomere elongation during ICM formation is controlled by mitochondrial-nuclear communication at fertilization. Using mitochondrially-targeted therapeutics (BGP-15, MitoQ, SS-31, metformin) we demonstrate that it is possible to modulate the pre-implantation telomere resetting process and restore deficiencies in neonatal telomere length.

Telomeres, the repetitive DNA caps at the ends of chromosomes, are crucial protectors of genome stability; and telomere length is a major determinant of healthy aging[1]. Telomeres shorten with each somatic cell division, and once below a threshold length, cellular senescence is triggered, contributing to tissue decline and pathology. Thus, long telomeres are associated with good health in old age and increased longevity, whereas shorter telomere length is a biomarker of aging and susceptibility to aging-associated co-morbidities, particularly cardiovascular disease[2,3]. Importantly, it is less well-recognized that telomere length at birth is a principal determinant of lifetime telomere length. It is more influential than environmental stressors that influence telomere attrition rate throughout the rest of life (reviewed in ref. 4).

Some babies are born with shorter telomeres than others, increasing their lifetime risk of chronic diseases associated with aging. In particular, shorter telomeres are observed in children of women with obesity or metabolic syndrome[5,6], or born to women of advanced reproductive age[7,8]. As adults, these individuals are at increased risk of premature mortality from cardiovascular events, even when they are not obese[9]. Children conceived through the use of in vitro embryo

[1]Robinson Research Institute, School of Biomedicine, The University of Adelaide, Adelaide, SA, Australia. [2]Genea Fertility SA, St. Andrews Hospital, Adelaide, SA, Australia. [3]Telomere Length Regulation Unit, Children's Medical Research Institute, Faculty of Medicine and Health, University of Sydney, Westmead, NSW, Australia. [4]Development and Stem Cells Program and Department of Anatomy and Developmental Biology, Monash Biomedicine Discovery Institute, Monash University, Melbourne, VIC, Australia. [5]Department of Biochemistry and Molecular Biology, Monash Biomedicine Discovery Institute, Monash University, Melbourne VIC, Australia. [6]Social Profit Network, Menlo Park, CA 94025, USA. [7]South Australian Health & Medical Research Institute, Adelaide, SA, Australia. [8]Monash Institute of Pharmaceutical Sciences, Monash University, Parkville, VIC, Australia. ✉e-mail: Rebecca.robker@adelaide.edu.au

culture have shorter telomeres than in vivo conceived peers[10]; and suboptimal vascular function is documented in similar cohorts at adolescence[11]. Yet despite shortened telomeres representing a major risk factor for chronic disease and early death, there is relatively limited understanding of the developmental mechanisms that establish neonatal telomere length, and that lead to shortened telomeres in some prenatal circumstances.

Preimplantation embryo development is an exceptional circumstance where telomeric DNA is extended. Specifically, rapid telomere elongation occurs in the first zygotic cell divisions between fertilization and blastocyst formation[12,13]. Telomere length achieved within the Inner Cell Mass (ICM), the pluripotent precursors to all fetal tissues, is considered to be the maximum set-point for that future individual[13,14]. Telomeres then continuously shorten at a relatively fixed rate during fetal development[15], childhood[16] and adulthood[17]. Thus, the initial elongation and resetting mechanism that establishes ICM telomere length is essential for genome integrity and determines progenitor cell replicative capacity in offspring.

We sought to identify the mechanisms that regulate resetting of ICM telomere length during embryogenesis, in order to understand how maternal aging and obesity might impact the set-point, and lead to shortened telomeres in offspring in these circumstances. Using mouse models, we find that mitochondrial-nuclear communication controls telomere elongation during pre-implantation embryogenesis; and show that this resetting is highly responsive to maternal physiology and in vitro environmental conditions. We also demonstrate that it is possible to modulate the telomere resetting process to restore deficiencies in neonatal telomere length.

## Results

### Telomere elongation between 8-cell and blastocyst formation is blunted by oxidative stress

Preimplantation development involves complete reprogramming of the parental nuclear chromatin between the time of fertilization and the formation of the ICM[18]; however the kinetics of telomere elongation are not well characterized. To comprehensively map telomere length dynamics across this phase of embryogenesis, mouse MII stage oocytes were fertilized in vitro and collected at precise developmental milestones between 2-cell (24 h post-fertilization) and blastocyst formation (96 h post-fertilization) (Fig. 1a). A quantitative PCR-based assay was developed to measure telomere DNA in individual oocytes and embryos normalized to a nuclear reference gene ($Rn18s$) to account for increases in cell number, and validated (Fig. S1). Telomere lengthening was detected by the 2-cell stage using both qPCR and quantitative telomere fluorescence in situ hybridization (FISH) (Fig. 1b, Fig. S2a) and further at the 8-cell stage (Fig. 1b), a process also observed by others using these same methodologies and thought to be recombination-mediated[12,13]. Between the 8-cell and blastocyst stages, telomere elongation was rapid (occurring within ~40 h) and of greater magnitude (Fig. 1b). This is consistent with the well-documented telomerase-mediated elongation that occurs between the morula to blastocyst stages[12–14]. Telomere length in isolated ICM was measured to verify telomere elongation in the embryo proper, compared to the trophectoderm (TE, the placental precursor), of individual micro-dissected blastocysts (Fig. 1c). This confirmed that telomere length is longer in the ICM (Fig. 1c;[14]). There was no further change in telomere length in immuno-purified ICM (see example Fig. 1d) between day 5 of culture (96 h post-fertilization) which corresponds to E4.5 (the time-frame of normal implantation) and day 6 of culture i.e., E5.5 in vivo (Fig. 1e). Telomere elongation during pre-implantation development was also examined in 4 additional strains of mice and consistently showed the greatest increase in telomere length only between 8-cells and blastocysts (Fig. S2b–e). These and previous observations[13,14] confirm that ICM telomere length in day 5 blastocysts is the maximum achieved during pre-implantation development.

Telomeres are GC-rich, which increases their susceptibility to accelerated attrition in response to elevated levels of intracellular Reactive Oxygen Species (ROS) in differentiated somatic cells[19]. In contrast, the potential impact of ROS-mediated oxidative stress on telomere elongation during preimplantation embryogenesis is not known. Thus, we induced oxidative stress during embryo culture and examined the impact on telomere length to better understand the regulation of telomere elongation kinetics. Mouse MII oocytes were exposed to 20% $O_2$ (similar to ambient air) during in vitro fertilization (IVF) and embryo culture (Fig. 1f), to induce oxidative stress. These were compared with embryos fertilized and cultured at 5% $O_2$ (which is the gold-standard for mouse and human in vitro embryo culture[20] because it better mimics the female reproductive tract). Telomere length in 8-cell embryos was not affected by culture in 20% $O_2$ (Fig. 1g), indicating no deficiency in the earliest phase of elongation, and no acceleration of telomere attrition. However, by the morula stage, embryos cultured in high oxygen exhibited shorter telomeres (Fig. 1h). The normal increase in telomere length per cell between the morula stage and ICM was blunted in embryos exposed to the oxidative 20% $O_2$ culture conditions (Fig. 1h). Validation in a larger cohort of embryos clearly demonstrated reduced telomeric DNA per cell in ICM from embryos cultured at high oxygen compared to those cultured in 5% $O_2$ (Fig. 1i). To confirm this result, identical embryos were assessed by quantitative telomere fluorescence in situ hybridization (FISH) of the dissociated ICM cells, which detects relative differences in telomere length (Fig. S2f;[13,14]). Shorter telomeres in the ICM of embryos cultured at 20% $O_2$ was reproduced using this second methodology (Fig. 1j). When these blastocyst stage embryos were transferred to uteri of surrogate females for gestation, those that had been exposed to 20% $O_2$ exhibited reduced survival (Fig. S4). Thus, telomere lengths in offspring could not be compared in equivalent fetal tissues, due to the confounding effect of fetal loss in one group. Overall, however, these results indicate a previously uncharacterized link between oxidative stress and telomere biology. Specifically, in contrast to the well documented mechanism of ROS-induced telomere shortening, we now show that oxidative stress impairs embryo telomere elongation, specifically at the morula and blastocyst stages of development.

Mitochondria are essential for cellular energy production but are also a major site of ROS production which is upregulated in response to cellular stressors[21–23]. During preimplantation development, mitochondria are also an intracellular source of metabolites and enzymes that influence embryo epigenetic reprogramming[24,25] and zygotic genome activation[26]. We therefore tested whether mitochondrial dysfunction may be a potential mechanistic link between oxidative stress and impaired telomere lengthening. Mitochondrial ROS (mtROS; reflective of mitochondrial superoxide production), tricarboxylic acid (TCA) cycle metabolites, and mitochondrial membrane potential (MMP; an indicator of oxidative phosphorylation) were measured to characterize the effects of 20% $O_2$-induced oxidative stress on mitochondrial bioenergetics (Fig. 2a). At 6 h following IVF, mtROS were increased in zygotes exposed to 20% $O_2$ (Fig. 2b). Metabolites of the TCA cycle were measured at this same developmental timepoint, to assess metabolic status. NAD(P)H levels were decreased in zygotes cultured in 20% $O_2$ (Fig. 2c, Fig. S5a), but FAD + + were not affected (Fig. 2d, Fig. S5b). These alterations in mitochondrial activity were not due to changes in mitochondrial mass since Mitotracker staining in zygotes (Fig. 2e, Fig. S5c) was not different between treatment groups. MMP, detected using TMRM potentiometric dye, was not altered in zygotes or 8-cell embryos, but the later marked increase in MMP seen at the morula and blastocyst stage was significantly blunted in embryos cultured in 20% $O_2$ (Fig. 2f).

To examine whether these disruptions to mitochondrial activity (increased mtROS, decreased NAD(P)H) are associated with zygotic nuclear changes, two different DNA epigenetic biomarkers were assessed in pronuclear stage zygotes (10 h post-IVF). High levels of

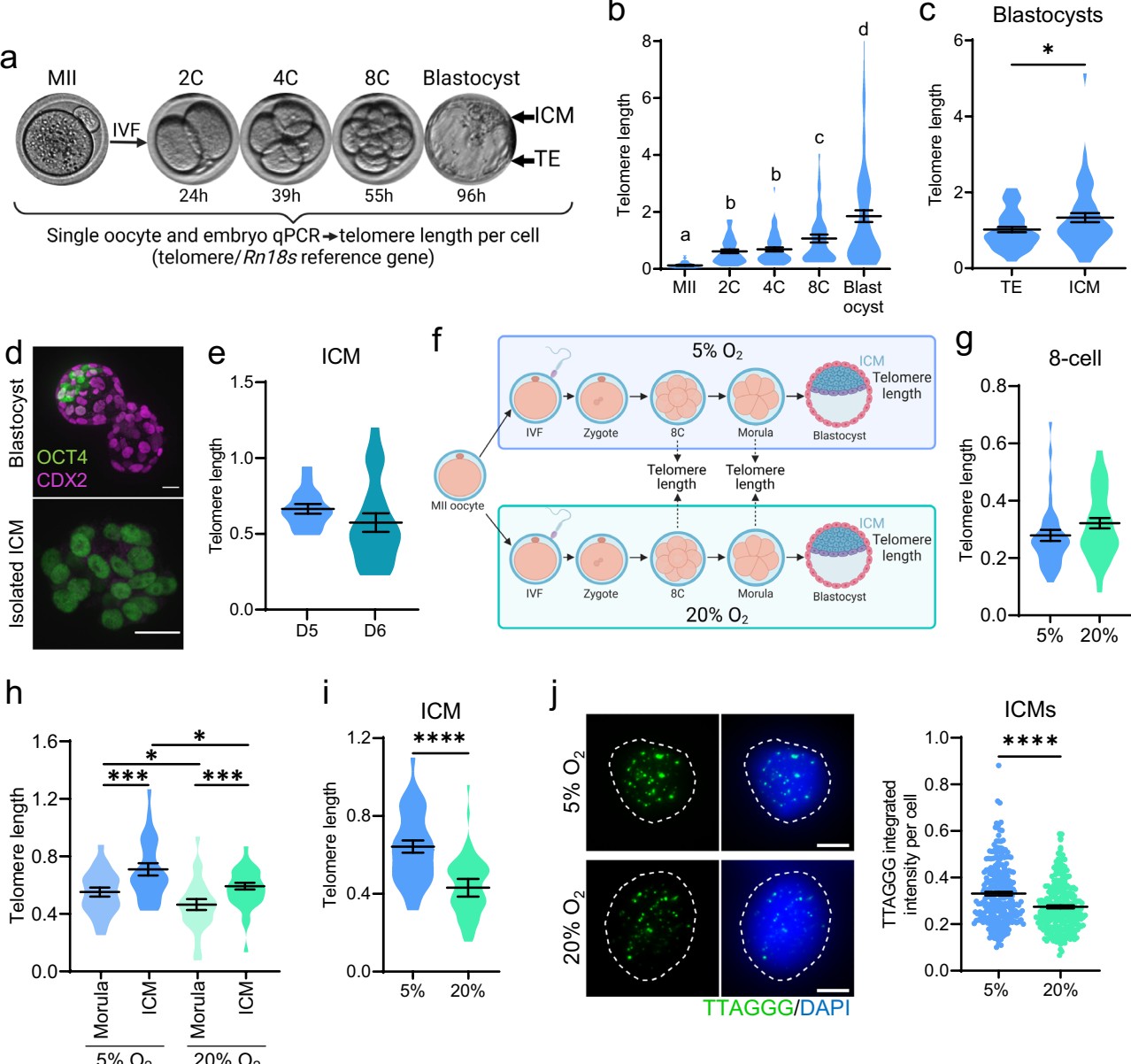

**Fig. 1 | Telomere elongation during pre-implantation embryogenesis is impaired by oxidative stress.** Mouse oocytes underwent in vitro fertilization (IVF) and embryos were collected at specific developmental stages at the times indicated (a) Created in BioRender. Gordon, Y. (2025) https://BioRender.com/u13t276. Telomere length (telomere/ *Rn18s* ($2^{-\Delta\Delta Ct}$)) was measured by qPCR. Telomere length in individual oocytes/ embryos (**b**; $n = 50$ MII, $n = 51$ 2 C, $n = 57$ 4 C, $n = 44$ 8 C, $n = 77$ blastocyst). Telomere length in dissected Inner Cell Mass (ICM) and Trophectoderm (TE) from the same blastocysts (**c**; $n = 51$). A whole blastocyst (**d**; upper) or isolated Inner Cell Mass (ICM) after immunosurgery (d; lower) to show purity of ICM cells (green, OCT4 + ) without TE cells (magenta, CDX2 + ). Telomere length in ICM from day 5 (D5, $n = 16$) and day 6 (D6, $n = 21$) blastocysts (**e**). MII oocytes were fertilized by IVF and cultured in vitro at either 5% or 20% oxygen (**f**) Created in BioRender. Gordon, Y. (2025) https://BioRender.com/x79j582. Telomere length in individual 8-cell embryos (**g**; $n = 36$ 5% O$_2$, $n = 47$ 20% O$_2$), morulae (**h**; $n = 24$ 5% O$_2$, $n = 25$ 20% O$_2$), and ICMs (h; $n = 25$ 5% O$_2$, $n = 28$ 20% O$_2$, **i**; $n = 16$ 5% O$_2$, $n = 18$ 20% O$_2$). Quantitative telomere FISH for telomere (green) with DAPI DNA stain (blue) in dissociated ICMs (j; left) and fluorescence per cell quantified (**j**; right; $n = 271$ nuclei from 30 ICMs derived from 4 mice per group, individual data points are plotted, horizontal lines are mean ± SEM. Violin plots show population distribution of telomere length, and horizontal lines are mean ± SEM (b, c, e, g-i). qPCR data was log transformed for statistical analysis. Statistical tests were: linear mixed-effects model (b); two-sided paired *t*-test (c), two-sided unpaired t-test (e, g, i, j); or two-way ANOVA (h). For (b) different letters indicate $p < 0.05$. For (c, h-j) *$p < 0.02$, ***$p = 0.0004$, ****$p < 0.0001$. For exact *P* values see Supplementary Table 1. Representative images shown, scale bar: 20 μm (d), 5 μm (j). Source data are provided as a Source Data File.

mtROS cause oxidative changes to DNA, with guanine bases most susceptible and converted to 8-oxo-7,8-dihydroguanine (8-oxodG)[27]. 8-oxodG immunostaining was visibly higher in the pronuclei of zygotes exposed to 20% O$_2$ (Fig. 2g). Secondly, hydroxymethylation of cytosine was measured by immunostaining as a general assessment of epigenetic reprogramming status[28]. Paternal pronuclei (the larger of the two) preferentially stained for 5-hydroxymethylcytosine (5hmC), the oxidized form of 5-methylcytosine (5mC) generated during Tet-mediated demethylation, as expected (Fig. 2h, Fig. S6). In embryos exposed to 20% O$_2$, the differential in 5hmC abundance between the paternal and maternal pronuclei was reduced (Fig. 2h, Supplemental Fig. S6) indicating that oxidative stress influences epigenetic marks on the zygotic nuclear DNA. Thus, the impaired telomere lengthening that occurs in response to 20% O$_2$-induced oxidative stress could be due to

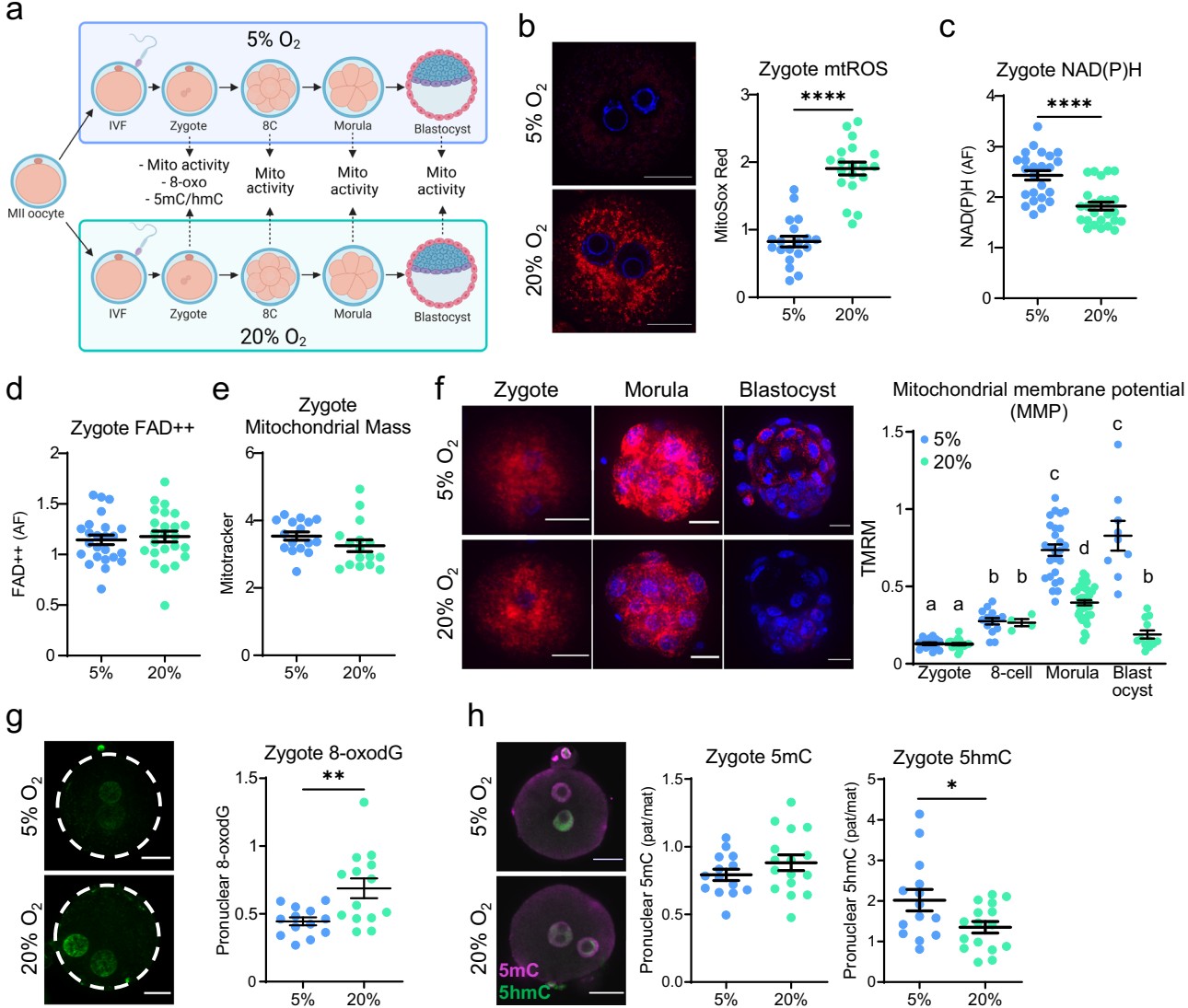

**Fig. 2 | High oxygen culture induces mitochondrial dysfunction and zygotic epigenetic alterations.** MII oocytes were fertilized by IVF and cultured in vitro at either 5% or 20% oxygen (**a**) Created in BioRender. Gordon, Y. (2025) https://BioRender.com/s35n498. Zygotes (6 h post-IVF) were labelled with mitochondrial superoxide (mtROS) indicator MitoSox Red (red, plus DNA stain Hoechst-3342 (blue)) and corrected total fluorescence determined (**b**; $n = 19$/group). Levels of NAD(P)H (**c**) and FAD++ (**d**) in zygotes 6 h post-IVF; $n = 24$/group. Mitochondrial mass was assessed in zygotes using MitoTracker Green (**e**; $n = 16$/group). Zygotes (4 h post-IVF), 8-cells, morulae and blastocysts were stained with mitochondrial membrane potential (MMP) indicator TMRM (red) plus Hoechst-3342 (blue) DNA stain (**f**; left) and fluorescence quantified (right; zygote: $n = 15$ 5% $O_2$, $n = 18$ 20% $O_2$, 8-cell: $n = 13$ 5% $O_2$, $n = 4$ 20% $O_2$, morula: $n = 26$ 5% $O_2$, $n = 41$ 20% $O_2$, blastocyst:

$n = 9$ 5% $O_2$, $n = 12$ 20% $O_2$). Zygotes (8 h post-IVF) were immuno-labeled with anti-8-oxodG (green) and fluorescence intensity in the pronuclei quantified (b; $n = 13$ 5% $O_2$; $n = 14$ 20% $O_2$). Zygotes (10 h post-IVF) were immuno-labeled with anti-5-methylcytosine (5mC; magenta) and anti-5-hydroxymethylcytosine (5hmC; green) antibodies (**h**; left). Fluorescence intensity was measured and expressed as the ratio in paternal versus maternal pronuclei (5% $O_2$: $n = 14$; 20% $O_2$: $n = 16$). Individual data points are plotted, horizontal lines are mean ± SEM (b-h). Statistical tests were: two-sided unpaired $t$-test (b-e, **g**, h); or one-way ANOVA (f). For (f) different letters indicate $p < 0.05$. For (b-e, g, h) *$p < 0.03$, **$p = 0.0056$, ****$p < 0.0001$. For exact $P$ values see Supplementary Table 2. Representative images shown, scale bar: 20 μm (b, f, g, h). Source data are provided as a Source Data File.

disruptions in either mitochondrial function and/or pronuclear reprogramming.

**Telomere length in offspring is regulated by oocyte mitochondria and zygotic nuclear reprogramming**

To directly test whether mitochondrial function regulates telomere elongation during preimplantation embryogenesis, we targeted mitochondrial bioenergetic activity using a highly specific Complex I inhibitor (rotenone). Initial experiments involved feeding female mice a low dose of rotenone (150 ppm in the diet, as in ref. 29) for three weeks prior to mating, at which time they were switched to the matched control diet; so that rotenone exposure was restricted to the pre-

conception period (Fig. 3a). Telomere lengths were then measured in fetal tissues (liver, kidney, heart and brain) collected at E18.5 which is near the time of birth (Fig. 3, Fig. S7). The fetuses of mothers exposed to rotenone prior to conception exhibited shorter telomeres in kidney and heart (Fig. 3b–d).

To restrict rotenone exposure (and mitochondrial disruption) more precisely to oocytes prior to fertilization only, female mice were fed rotenone and then ovulated oocytes underwent IVF under identical conditions as controls, followed by uterine transfer at the blastocyst stage (Fig. 4a). At 6 h post-IVF, zygotes from oocytes of rotenone-exposed females exhibited markedly elevated mtROS (Fig. 4b). Further, rotenone-exposed zygotes displayed increased levels of NAD(P)H

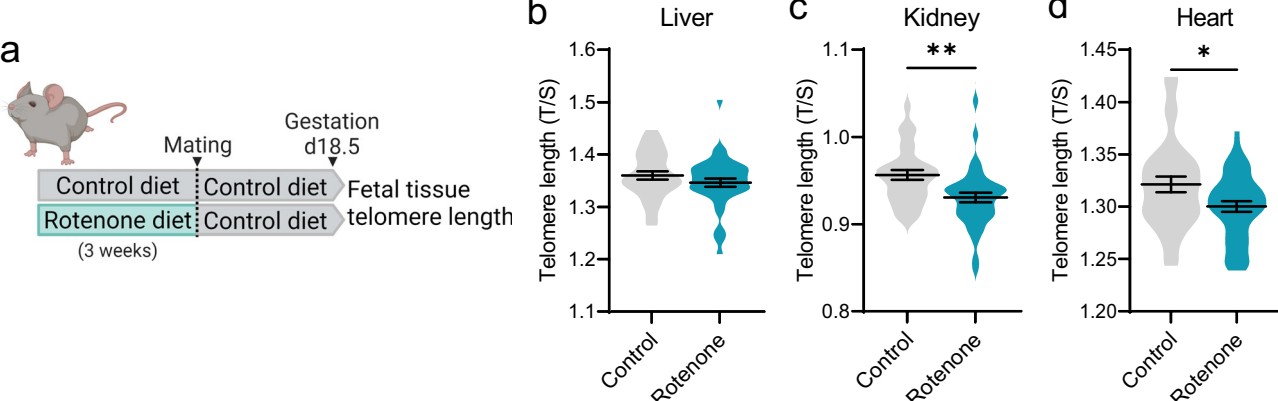

**Fig. 3 | Pre-conception rotenone exposure reduces fetal telomere length.** Mice were exposed to rotenone (150ppm in chow) for three weeks before mating (**a**), and fetal tissues collected at day 18.5 of pregnancy for assessment of telomere length. Created in BioRender. Gordon, Y. (2025) https://BioRender.com/b05w439. Relative telomere length (telomere/ *Rn18s* ($2^{-\Delta\Delta Ct}$)) was analyzed in liver (**b**), kidney (**c**), and heart (**d**) of $n = 33$ control and $n = 40$ rotenone fetuses. Violin plots show population distribution, and horizontal lines are mean ± SEM. qPCR data was log transformed for statistical analysis via two-sided unpaired *t*-test. *$p = 0.0206$, **$p = 0.0016$. For exact *P* values see Supplementary Table 3. Source data are provided as a Source Data File.

(Fig. 4c, Fig. S5d) and FAD + + (Fig. 4d, Fig. S5e) compared to controls, despite having similar mitochondrial mass, as indicated by Mito-Tracker (Fig. 4e, Fig. S5f). Oxidative DNA lesions (8-oxodG) were increased in zygotes from oocytes of rotenone-exposed females (Fig. 4f). Disrupted 5mC/5hmC epigenetic patterning was also apparent in zygotes from oocytes of mice that were fed rotenone (Fig. 4g, Fig. S8). Rotenone-induced disruption to MMP was present in zygotes and persisted in 8-cell embryos (Fig. 4h). Taken together, these results indicate that rotenone-induced mitochondrial dysfunction in oocytes leads to an induction of mtROS and oxidative DNA damage, as well as alterations to mitochondrial metabolism and pronuclear reprogramming. The large induction of mtROS was the only mitochondrial phenotype that was similar between zygotes conceived in 20% $O_2$ and those conceived from rotenone-exposed oocytes.

Telomere length was shorter in MII oocytes from rotenone-exposed females compared to controls (Fig. 4i), likely due to ROS-mediated telomere attrition during folliculogenesis. This deficit was, however, reversed by the 8-cell stage (Fig. 4i), again suggesting that the earliest phase of telomere elongation is not influenced by mitochondrial bioenergetics or oxidative stress. Although some telomere lengthening occurred between the 8-cell and blastocyst stages in embryos from oocytes of rotenone-exposed females, blastocyst telomeres were significantly shorter (Fig. 4i), specifically within the ICM (Fig. 4j). Shorter telomeres in the ICM of embryos conceived from rotenone-exposed oocytes was confirmed via quantitative telomere FISH in an independent cohort of blastocysts (Fig. 4k). These results demonstrate that direct perturbation of oocyte mitochondrial function prior to fertilization impairs later telomere elongation capacity, specifically between the 8-cell and blastocyst stages of pre-implantation development.

Blastocysts derived from oocytes of control or rotenone-exposed females were transferred to uteri of surrogates (to provide identical gestational environments) and fetuses examined near the time of birth to determine whether the deficiency in ICM telomere length persisted in offspring (see Fig. S9). Telomeres were shorter in the hearts of offspring (Fig. 4l), mirroring the relative differences in the ICM. These data demonstrate that relative differences in telomere length in the pluripotent ICM cells of the blastocyst persist through fetal development at least in some tissues.

Because rotenone-induced mitochondrial dysfunction caused impaired telomere elongation, we investigated whether it was possible to restore ICM telomere length through improved oocyte mitochondrial activity. We utilized BGP-15, a hydroximic acid compound which improves mitochondrial bioenergetics[30,31] and prevents fragmentation[32] in contexts of induced oxidative stress, and exhibits therapeutic utility in pre-clinical pathologies involving mitochondrial dysfunction[33–35]. Mice fed rotenone were treated with BGP-15 (100 mg/kg by i.p. injection) or saline vehicle for 4 days prior to gonadotropin-stimulated ovulation (as in refs. 33,34) and IVF under identical conditions (Fig. 5a). This pre-conception BGP-15 treatment reduced mtROS levels in zygotes of rotenone-fed mice (Fig. 5b). BGP-15 treatment of rotenone-fed females also mitigated the later defect in ICM telomere length, and they were equivalent to those of the controls (Fig. 5c).

To further interrogate the developmental timeframes that influence telomere elongation capacity, we tested whether direct acute reduction of mtROS at fertilization, via in vitro administration of BGP-15, could restore blastocyst telomere lengthening. When oocytes from control or rotenone-exposed mice were treated with BGP-15 (10 μM) in the IVF and embryo culture media (Fig. 5d), mtROS levels were reduced within 6 h in zygotes from oocytes of rotenone-exposed mice (Fig. 5e). In vitro treatment of oocytes and embryos with BGP-15 also reversed the deficit in ICM telomere length (Fig. 5f). Cumulatively, these data indicate that ICM telomere length is tightly coupled to oocyte mtROS levels at fertilization, and that this biology is modifiable to influence the telomere set-point.

To further investigate the role of mitochondrial activity during the window between fertilization and pronuclear formation on subsequent telomere length in ICM, ovulated MII oocytes from rotenone-fed mice (or controls) were treated with BGP-15 or other mitochondrially-targeted compounds for only 8 h from IVF to PN formation (Fig. 6a). Treatment of rotenone-exposed eggs with BGP-15 for 8 h from the time of fertilization reduced mtROS in zygotes (Fig. 6b) as expected; and even though these zygotes were transferred to fresh media that did not contain BGP-15, they exhibited improved telomere elongation in the ICM (Fig. 6c). A similar experiment treated rotenone-exposed eggs with MitoQ, a well characterized mitochondria-targeted CoQ10 ubiquinone[36], for just 8 h from the time of fertilization (Fig. 6a). MitoQ treatment of rotenone-exposed eggs reduced mtROS levels in zygotes (Fig. 6d); and following culture in standard media until the blastocyst stage, led to increased telomere length in ICM (Fig. 6e). Lastly, rotenone-exposed eggs were treated in exactly the same manner with SS-31 (in clinical development as Elamipretide), a synthetic tetrapeptide that binds cardiolipin within the mitochondrial inner membrane to improve cellular bioenergetics[37,38]. SS-31 treatment of rotenone-exposed eggs reduced mtROS levels in zygotes (Fig. 6f); and

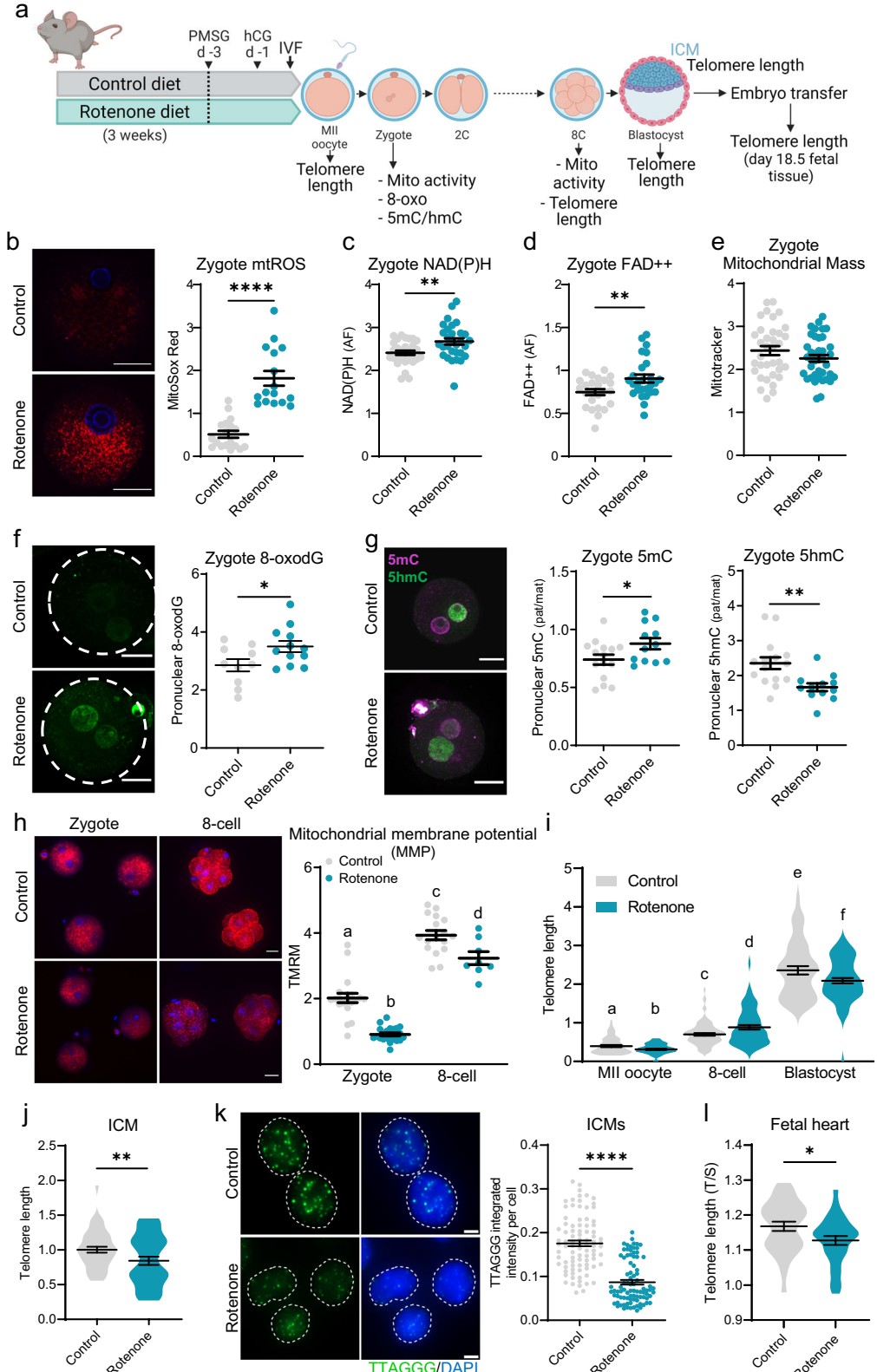

led to increased telomere length in ICM (Fig. 6g). Cumulatively, these results demonstrate that mtROS levels at the time of fertilization tightly correspond to telomere length in ICM. Specifically, three different compounds that are known to act on mitochondria and which normalize mtROS in rotenone-exposed zygotes, improved telomere elongation capacity in the ICM even when present only during the first cell cycle.

To dissect the roles of mitochondria versus nuclear events in regulating telomere elongation in ICM, we conducted reciprocal pronuclear transfers. At the 1-cell stage (~8 h post-fertilization), pronuclei from zygotes of either rotenone-fed or control mice were fused with enucleated cytoplasts of rotenone-exposed or control zygotes, resulting in four types of reconstructed embryos which were then cultured to the blastocyst stage (Fig. 6h; Fig. S10). ICM of embryos

**Fig. 4 | Oocyte mitochondrial dysfunction causes pronuclear DNA modifications and reduces embryo and fetal telomere length.** Mice were exposed to rotenone (150 ppm in chow) for three weeks before ovulation, IVF and assessments of embryos and fetuses (**a**) Created in BioRender. Gordon, Y. (2025) https://BioRender.com/u42e503. Zygotes were labelled with MitoSox Red (red) and DNA stain Hoechst-3342 (blue) and fluorescence measured (**b**; $n = 18$ control, $n = 16$ rotenone). Levels of NAD(P)H (**c**) and FAD++ (**d**) in zygotes 6 h post-IVF; $n = 30$/group. Mitochondrial mass in zygotes was assessed using MitoTracker Green (**e**; $n = 35$ control, $n = 39$ rotenone). Zygotes (8 h post-IVF) were immuno-labeled with anti-8-oxodG (green) and fluorescence intensity in–– pronuclei quantified (**f**; $n = 10$ control, $n = 12$ rotenone). Zygotes were co-stained with anti-5-methylcytosine (5mC; magenta) and anti-5-hydroxymethylcytosine (5hmC; green) antibodies (**g**). Fluorescent signal intensity was quantified for 5mC (left) and 5hmC (right) in maternal and paternal pronuclei ($n = 15$ control, $n = 12$ rotenone) and presented as the ratio in paternal versus maternal pronuclei. Zygotes and 8-cell embryos were labelled with MMP indicator TMRM (red, and Hoechst-3342 (blue)) and red fluorescence measured (**h**; zygote: $n = 20$ control, $n = 22$ rotenone; 8-cell $n = 17$ control, $n = 8$ rotenone). Telomere length (telomere/$Rn18s$ ($2^{-\Delta\Delta Ct}$)) in individual MII oocytes, 8-cells, blastocysts (**i**; MII oocyte: $n = 63$ control, $n = 66$ rotenone; 8-cell: $n = 86$ control, $n = 61$ rotenone, blastocyst: $n = 45$ control, $n = 69$ rotenone), and ICMs (**j**; $n = 42$ control, $n = 38$ rotenone). Quantitative telomere FISH for telomere (green) with DAPI DNA stain (blue) in dissociated ICMs (**k**; left) and fluorescence per cell quantified (right; $n = 88$ nuclei from 33 ICMs derived from 3 mice per group). IVF-conceived blastocysts from rotenone-treated (or control) mice were transferred to surrogate females for gestation, and fetal tissues collected at day 18.5 of pregnancy for qPCR telomere length analysis of heart (**l**; $n = 36$ control, $n = 35$ rotenone fetuses). Individual data points are plotted, horizontal lines are mean ± SEM (b-h, k). Violin plots show population distribution, and horizontal lines are mean ± SEM (i, j, l). qPCR data was log transformed for statistical analysis. Statistical tests were two-sided unpaired $t$-test (b-g, j-l), one-way ANOVA (h) or two-sided unpaired $t$-test between the same developmental stage and one-way ANOVA between different developmental stages of the same group (i). For (h, i) different letters indicate $p < 0.05$. For (b-g, j-l) *$p < 0.05$, **$p \le 0.009$, ****$p < 0.0001$. For exact $P$ values see Supplementary Table 4. Representative images shown, scale bar: 20 µm (b, f, g, h), 2 µm (k). Source data are provided as a Source Data File.

reconstructed from rotenone-exposed pronuclei plus rotenone-exposed cytoplast had shorter telomeres than those of embryos reconstructed from components of controls (Fig. 6i), consistent with the relative differences in non-reconstructed embryos (Fig. 4j, k). Interestingly, telomere length was also reduced in embryos reconstructed from pronuclei of rotenone-exposed zygotes (Fig. 6i). This indicates that the mitochondrial damage present in rotenone-exposed zygotes altered the nuclear DNA in a manner that resulted in impaired telomere elongation in ICM. Thus, the nuclear material of rotenone-exposed zygotes contains the molecular signals leading to later impaired telomere elongation potential. Conversely, reconstructed embryos that received cytoplasm of rotenone-exposed zygotes did not have reduced telomere length in ICM (Fig. 6i). This indicates that the signaling from damaged mitochondria to nuclear DNA, that impairs telomere elongation, occurs prior to syngamy; and that the presence of damaged mitochondria following the pronuclear stage does not impact this process. Thus, disrupted mitochondrial function within the cytoplasm of the oocyte at fertilization causes pronuclear epigenetic changes prior to syngamy that are responsible for impaired telomere elongation in ICM at the blastocyst stage.

**Maternal aging or obesity impairs telomere elongation in ICM that is reversible by restoring oocyte mitochondrial function**

Oocyte mitochondrial function is known to be negatively impacted in the physiological contexts of aging[39,40] and obesity[34,41]. We investigated whether preimplantation telomere elongation is impacted in these circumstances and reversible using BGP-15 or other compounds to restore mitochondrial function. Female mice that were reproductively aged (12 months old, which extrapolates to 38-45 years old in women[42]) or young (3-4 months old) were treated with BGP-15 (100 mg/kg/day for 4 days) or vehicle, and ovulated oocytes underwent IVF under identical conditions (Fig. 7a). Ovulated oocytes of the aged mice exhibited high levels of mtROS that were reduced in females treated with BGP-15 (Fig. 7b). Nuclear changes were also observed with zygotes of aged females exhibiting an increase in 8-oxodG (Fig. S11a). Zygotes from aged females also exhibited a trend towards altered relative levels of 5mC and 5hmC in paternal versus maternal pronuclei; while in those given pre-conception treatment with BGP-15, ratios were the same as the zygotes of young mice (Fig. S11b–g). At the morula stage, mitochondrial membrane potential (MMP) was reduced in morulae from the older mice; and this deficit was normalized in embryos of females given pre-conception BGP-15 treatment (Fig. 7c). These mitochondrial phenotypes are identical to those of embryos exposed to 20% O$_2$ or from rotenone-fed mice.

Ovulated (MII) oocytes from old mice had shorter telomeres (Fig. 7d), and interestingly, oocyte telomere length was shorter in nulliparous females (that experienced continuous ovarian cycling) compared to littermates that were maintained in breeding pairs and continuously pregnant (Fig. 7e). Thus, in animals of the exact same age, limiting the number of estrous cycles appears to have a protective effect on oocyte telomere length. Although MII oocytes from older mice have shorter telomeres, following IVF, 8-cell embryos from old and young mice had no differences in telomere length (Fig. 7f); indicating a normalization of the defect in oocytes. However, by the blastocyst stage, those derived from aged mice had shorter telomeres than those from young mice (Fig. 7g) indicating a deficiency in the later phase of lengthening. Remarkably, BGP-15 treatment of aged mice (for 4 days prior to ovulation) restored telomere length in blastocysts (Fig. 7g). Telomere length in the ICM population mirrored that in whole blastocysts, with maternal reproductive aging leading to reduced ICM telomere length, and BGP-15 treatment restoring this deficit (Fig. 7h).

Given the notable effects of BGP-15 to restore ICM telomere length, two other pharmaceutical agents characterized to improve mitochondrial bioenergetics were tested. MitoQ[36] is available over-the-counter; while, metformin[43] is commonly prescribed for the treatment of insulin resistance, including in women seeking pregnancy. Reproductively old (12 months) female mice were administered either metformin (2 mg/mL) or MitoQ (150 µM) in their drinking water for two weeks prior to gonadotropin-stimulated ovulation and IVF under standard conditions (Fig. 7i). Treatment of aged females with metformin or MitoQ reduced mtROS levels within 6 h of fertilization (Fig. 7j). Metformin (but interestingly not MitoQ which was effective in vitro) restored ICM telomere length, such that it was indistinguishable from embryos derived from young females (Fig. 7k). These results further substantiate that telomere elongation capacity during preimplantation development is regulated by maternal factors contained within the ovulated oocyte, particularly mitochondrial activity; and provide proof-of-concept that this biology can be targeted in physiological contexts and using currently available drugs to influence embryo telomere length.

The impact of maternal obesity on offspring telomere elongation kinetics and the capacity to reverse any defects was examined in mice that are obese due to hyperphagia[34]. Direct acute manipulation of mitochondrial bioenergetics at fertilization was achieved through in vitro administration of BGP-15 (10µM) or SS-31 (1 nM) in the IVF and embryo culture media (Fig. 8a). When oocytes from lean control or obese mice were treated with BGP-15, mtROS levels were reduced within 6 h in zygotes from oocytes of obese mice (Fig. 8b), an identical response as observed in zygotes generated from oocytes of rotenone-fed mice (Fig. 5e, Fig. 6b), substantiating the efficacy of BGP-15 to directly influence mtROS production. Treatment of oocytes from

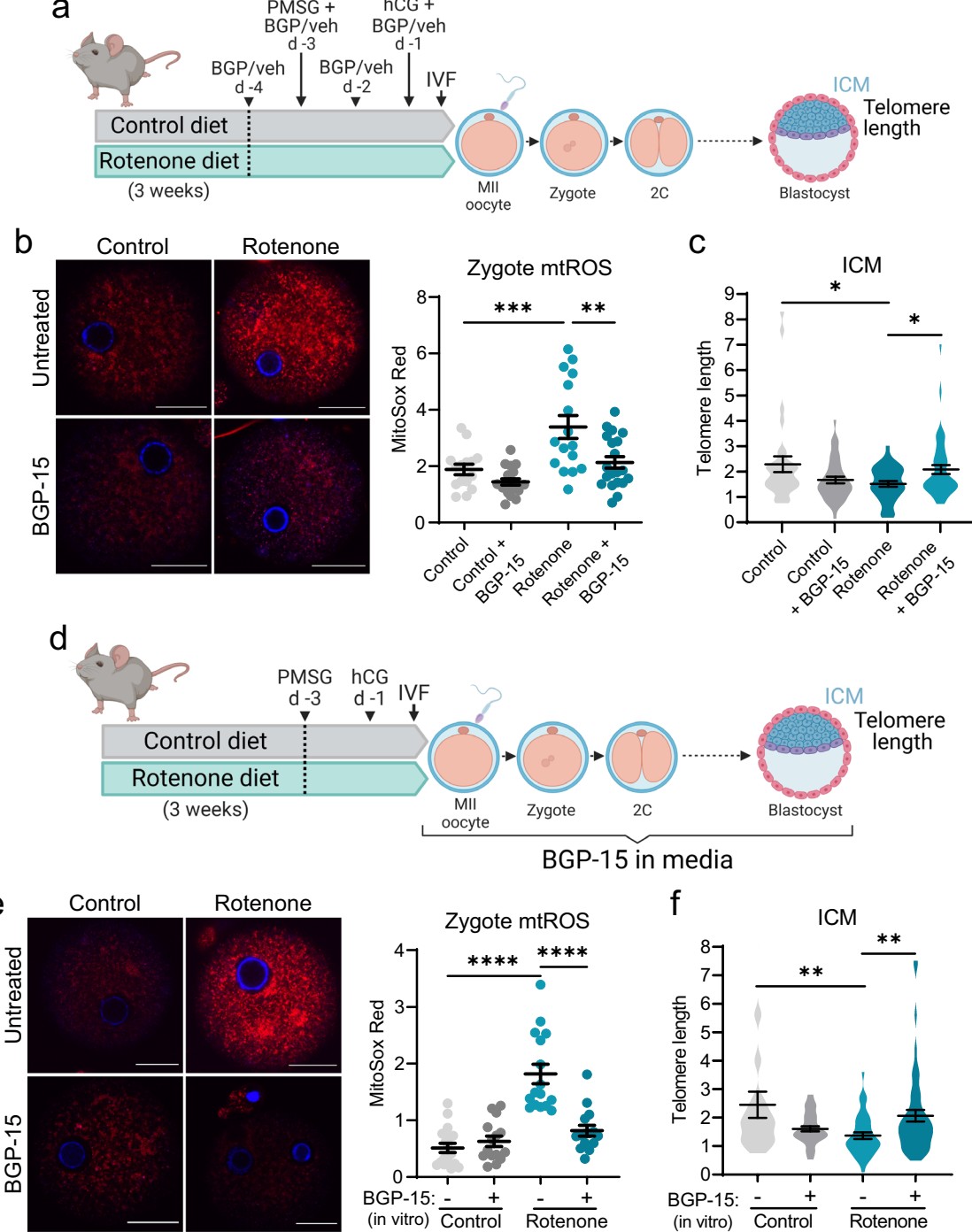

**Fig. 5 | BGP-15 reduces mtROS in rotenone-exposed zygotes and restores ICM telomere length.** Mice fed rotenone were treated with BGP-15 (100 mg/kg by i.p. injection, or saline vehicle) for 4 days prior to gonadotropin-stimulated ovulation and IVF (**a**) Created in BioRender. Gordon, Y. (2025) https://BioRender.com/d94e058. MitoSox Red (red, with DNA stain Hoechst-3342; blue) in zygotes 6 h post-fertilization (**b**; $n = 15$ control, $n = 19$ control+BGP-15, $n = 16$ rotenone, $n = 20$ rotenone+BGP-15). Telomere length (telomere/ *Rn18s* ($2^{-\Delta\Delta Ct}$)) in isolated ICMs (**c**; $n = 31$ control, $n = 40$ control+BGP-15, $n = 38$ rotenone, $n = 44$ rotenone+BGP-15). Oocytes derived from control or rotenone-exposed females underwent in vitro fertilization in media containing 10 μM BGP-15 (+) or media with the equivalent volume of vehicle (-) (**d**) Created in BioRender. Gordon, Y. (2025) https://

BioRender.com/t69m396. At 6 h post-fertilization, zygotes were labelled with MitoSox Red (red, and DNA stain Hoechst-3342; blue) and red fluorescence measured (**e**; $n = 18$ control, $n = 15$ control+BGP-15, $n = 16$ rotenone, $n = 15$ rotenone +BGP-15). Telomere length (telomere/ *Rn18s* ($2^{-\Delta\Delta Ct}$)) in isolated ICMs (**f**; $n = 34$ control, $n = 41$ control +BGP-15, $n = 39$ rotenone, $n = 53$ rotenone+BGP-15). Individual data points are plotted, horizontal lines are mean ± SEM (b, e). Violin plots show population distribution, and horizontal lines are mean ± SEM (c, f). qPCR data was log transformed for statistical analysis via one-way ANOVA; \*$p < 0.03$, \*\*$p \leq 0.0099$, \*\*\*$p = 0.0006$, \*\*\*\*$p < 0.0001$. For exact $P$ values see Supplementary Table 5. Representative images shown, scale bar: 20 μm (b, e). Source data are provided as a Source Data File.

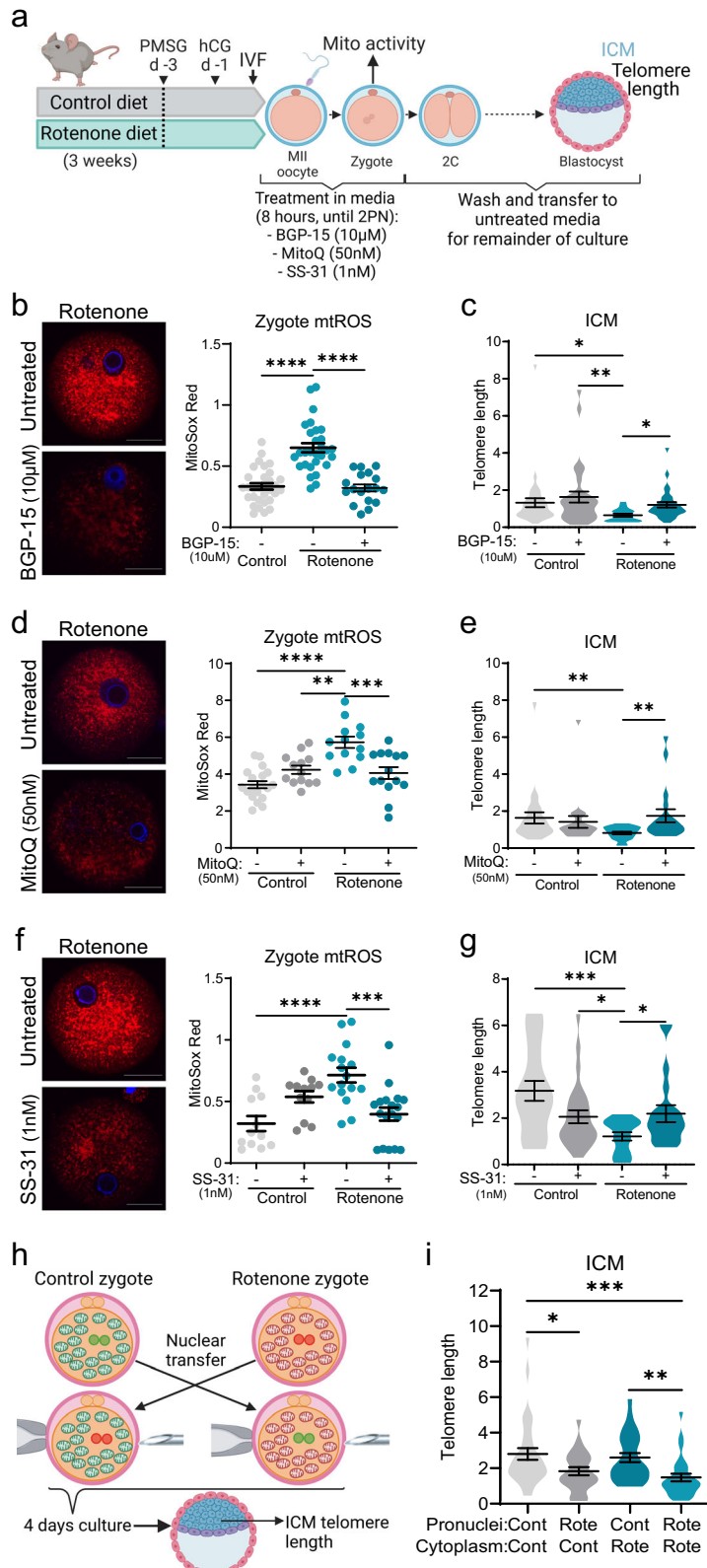

obese mice with BGP-15 during IVF and embryo culture also restored MMP at the morula stage (Fig. 8c). Most notably, this in vitro BGP-15 treatment normalized the deficiency in ICM telomere length in blastocysts (Fig. 8d). Similar results were achieved when ovulated eggs were treated with SS-31 during IVF and embryo culture: eggs from obese mice that were fertilized and cultured in SS-31 had reduced mtROS at the zygote stage (Fig. 8e) and increased ICM telomere length

in blastocysts (Fig. 8f). These results further substantiate that telomere elongation capacity can be acutely regulated even from the time of fertilization, and in a physiological context such as obesity.

Subsequent experiments examined the effects of maternal obesity on offspring telomere length and the capacity for reversal of any defects in mice treated for 4 days prior to ovulation with BGP-15. Ovulated oocytes from all groups underwent IVF under identical

**Fig. 6 | Mitochondria-nuclear communication at fertilization determines ICM telomere length.** Ovulated eggs from control or rotenone-exposed females underwent in vitro fertilization and embryo culture in media containing 10 μM BGP-15, 50 nM MitoQ, or 1 nM SS-31 (+) or media with the equivalent volume of vehicle (-) for 8 h and zygotes were then cultured in standard (untreated) media to the blastocyst stage (**a**) Created in BioRender. Gordon, Y. (2025) https://BioRender.com/y04d797. At 6 h post-fertilization, zygotes were labelled with MitoSox Red superoxide (mtROS) indicator (red, and DNA stain Hoechst-3342; blue) and red fluorescence measured (**b**: $n = 31$ control, $n = 29$ rotenone, $n = 18$ rotenone+BGP-15; **d**: $n = 19$ control, $n = 13$ control+MitoQ, $n = 13$ rotenone, $n = 14$ rotenone+MitoQ; **f**: $n = 12$ control, $n = 12$ control+SS-31, $n = 16$ rotenone, $n = 18$ rotenone+SS-31. At the blastocyst stage ICM telomere length (telomere/ $Rn18s$ ($2^{-\Delta\Delta Ct}$)) was measured by qPCR (**c**: $n = 35$ control, $n = 31$ control+BGP-15, $n = 18$ rotenone, $n = 30$ rotenone +BGP-15; **e**: $n = 24$ control, $n = 19$ control+MitoQ, $n = 21$ rotenone, $n = 18$ rotenone +MitoQ; **g**: $n = 19$ control, $n = 26$ control+SS-31, $n = 16$ rotenone, $n = 19$ rotenone+SS-

31). Mice were exposed to rotenone (150 ppm in chow) for three weeks before hormone stimulation, mating and zygote collection. Pronuclei were transferred between zygotes derived from control (cont) or rotenone-exposed (rote) females in each possible combination, and reconstructed embryos cultured to blastocyst (**h**) Created in BioRender. Gordon, Y. (2025) https://BioRender.com/h97n093. ICMs from blastocysts were analyzed for telomere length (telomere/ $Rn18s$ ($2^{-\Delta\Delta Ct}$)) (**i**: $n = 33$ cont-cont, $n = 24$ rote-cont, $n = 27$ cont-rote, $n = 28$ rote-rote). Individual data points are plotted, horizontal lines are mean ± SEM (**b, d, f**). Violin plots show population distribution, and horizontal lines are mean ± SEM (**c, e, g, i**). qPCR data was log transformed for statistical analysis via two-way ANOVA (**c, e, g**) or one-way ANOVA (**i**), MitoSox Red analyzed via one-way ANOVA (**b, d, f**); *$p < 0.05$, **$p < 0.008$, ***$p \leq 0.0004$, ****$p < 0.0001$. For exact $P$ values see Supplementary Table 6. Representative images shown, scale bar: 20 μm (**b, d, f**). Source data are provided as a Source Data File.

conditions (Fig. 9a). High mtROS (Fig. 9b) and altered 5mC/5hmC pronuclear epigenetic patterns (Fig. S12) were observed in zygotes of obese mice and normalized by pre-conception treatment with BGP-15. Mitochondrial membrane potential (MMP) was reduced in 8-cell embryos and morulae and this deficit was normalized in embryos of females given pre-conception BGP-15 treatment (Fig. 9c).

Telomere length was reduced in MII oocytes from obese mice when compared to lean, but not influenced by pre-conception treatment with BGP-15 (Fig. 9d). At the 8- cell stage, telomere length was not different between any of the groups (Fig. 9e) similar to the normalization that occurred in embryos exposed to other mitochondrial stressors (i.e. 20% $O_2$, rotenone, aging). At the blastocyst stage, telomeres were shorter in embryos derived from obese mice compared to those from lean controls (Fig. 9f), a deficiency apparent in the ICM (Fig. 9g). BGP-15 treatment of obese mice restored blastocyst telomere length (Fig. 9f), including in the ICM (Fig. 9g), such that it was similar to lean controls.

Treatment of obese females with metformin (2 mg/mL) or MitoQ (150 μM) in their drinking water for two weeks prior to gonadotropin-stimulated ovulation also reduced mtROS levels by 6 h post-fertilization (Fig. 9h). Both treatments resulted in restored ICM telomere length, to levels similar to embryos derived from lean mice (Fig. 9i). This interesting observation that embryos of obese mice were improved following treatment with either metformin or MitoQ, while those of aged mice were only improved by metformin, suggests distinct cellular deficiencies in oocytes of obese versus aged females, or differential sensitivity to MitoQ. Cumulatively, however these data demonstrate that physiological disruptions to oocyte mitochondria (via aging or obesity) lead to impaired telomere elongation and reduced telomere length in the ICM of offspring; but that this defect is reversible with pharmaceutical treatment.

Blastocysts derived from oocytes of lean mice, obese mice, and obese mice treated with BGP-15 (in vivo for 4 days) were transferred to identical lean surrogates and telomere length analyzed in day 14.5 fetal tissues. Telomere length was shorter in liver, kidney and heart of fetuses derived from oocytes of obese females (Fig. 9j), demonstrating that relative differences in the ICM (Fig. 9g) were maintained through fetal development. Further, fetuses from oocytes of obese females treated with pre-conception BGP-15 had increased telomere length in both heart and liver compared to offspring from untreated obese controls (Fig. 9j). Thus, maternal obesity leads to fundamental defects in the oocyte that impair telomere elongation in the ICM, a deficiency in the telomere set-point that persists through gestation. This molecular change is preventable by systemic treatment with a mitochondria-targeted therapeutic in the days prior to ovulation.

## Discussion

This work reveals new information about the mechanisms that control telomere regeneration during early life and the resetting of offspring

telomere length. Specifically, between fertilization and the first zygotic cell division, mitochondrial function impacts epigenetic remodeling to regulate telomere elongation during the second half of preimplantation development, thus determining the ICM telomere set-point. Via this process, relative differences in telomere length at birth, a major determinant of lifetime health and longevity, are established within the first few days following fertilization. These findings have important clinical and health implications. By demonstrating that maternal obesity and age impair embryo telomere elongation, they provide a potential mechanistic explanation for the occurrence of shorter telomeres in some infants[5–8]. Reassuringly, common assisted reproduction techniques did not influence telomere elongation in our model. Specifically, ICM telomere length was not altered by IVF and culture in optimized conditions compared to in vivo–conceived embryos (Fig. S3b, c), by extended embryo culture to day 6 (Fig. 1e), or by vitrification at the morula stage (Fig. S4a, S9a). However, by demonstrating that high oxygen culture impairs embryo telomere elongation, we identify oxidative stress during culture to blastocyst as a possible basis for shorter telomeres in IVF-conceived babies[10]. Importantly, these changes in telomere length are distinct from those caused by rare genetic disorders that result in profound telomere length defects and clinical syndromes[44,45] and instead provide a potential mechanistic basis for the natural variations in telomere length that are present in humans at birth. More broadly, these findings substantiate that maternal health and environmental conditions at the time of conception have long term consequences[46]: that now include influencing offspring susceptibility (or resilience) to aging and aging-associated diseases in later life.

Previously described links between mitochondria and telomere biology largely revolve around ROS production accelerating telomere attrition. For instance, severe maternal stress during pregnancy causes oxidative cellular damage and advanced telomere shortening in offspring[47], including in humans[48]. However, we now identify a different regulatory communication between the two organelles; wherein optimal mitochondrial function in the zygote is required for normal telomere elongation capacity, during the developmental window between 8-cells and ICM formation. Appropriate regulation of mitochondrial metabolism is also central to establishing embryonic stem cell pluripotency[49] and subsequent cell fate acquisition[50]; while telomere elongation is a hallmark and requisite component of nuclear reprogramming that is also linked to stemness[51,52]. Thus, our identification of this connection between mitochondria and telomeres during preimplantation embryogenesis may also extend to other contexts of cellular reprogramming and stem cell biology. In particular, this mechanism controlled by oocyte mitochondria may explain the enhanced telomere rejuvenation that occurs in embryonic stem cells generated by SCNT (somatic cell nuclear transfer) compared to iPSCs (induced pluripotent stem cells)[53].

Telomere length was measured by qPCR because it is the only technique currently applicable to the small amounts of DNA available

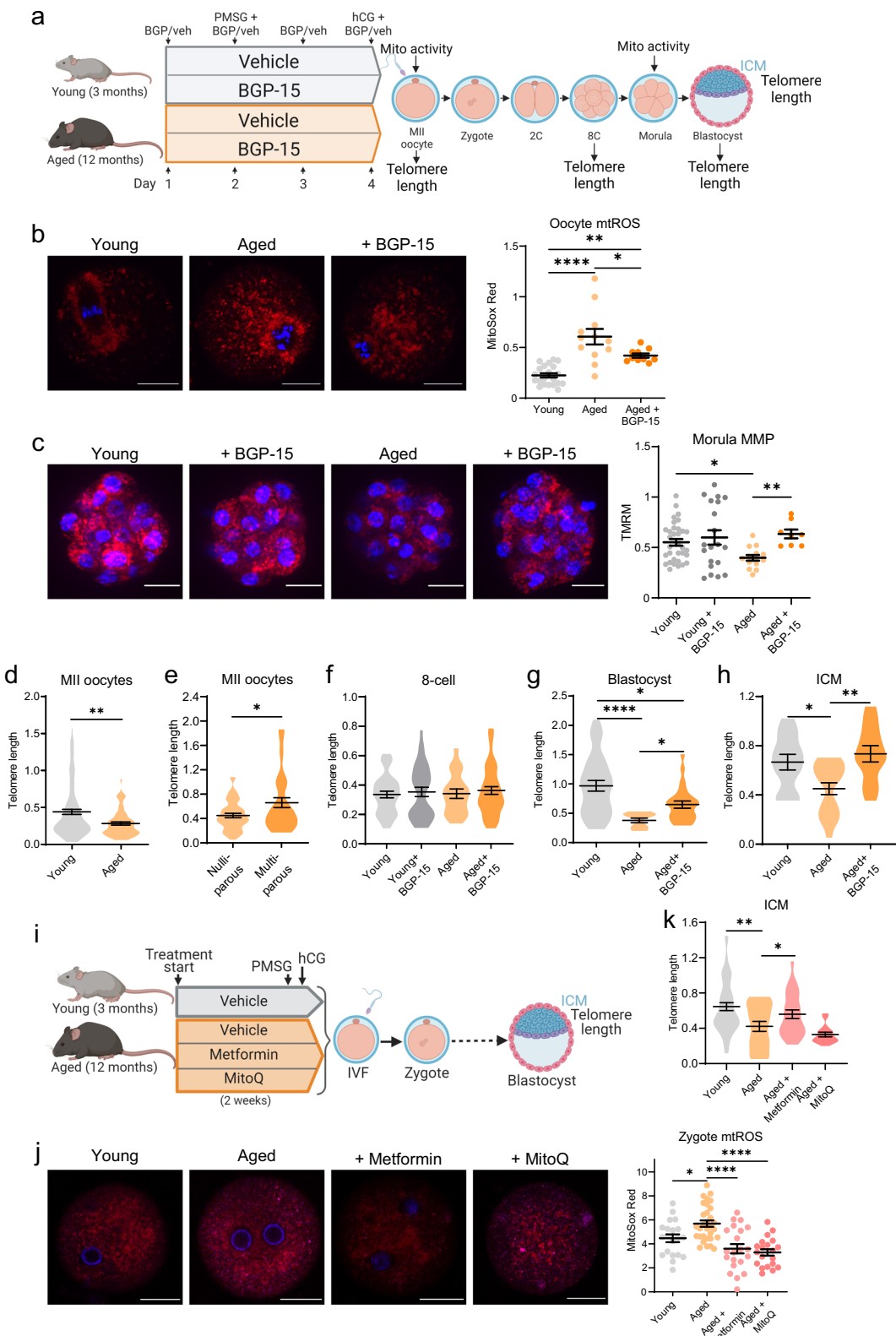

from single oocytes and embryos. We developed an in-house qPCR assay for relative telomere length measurement, which was extensively validated (Fig. S1, Fig. S2) and avoids PCR artefacts from non-specific primer binding and pre-PCR amplification steps. We further used quantitative telomere FISH which we and others[12–14] have previously established to measure mouse ICM telomere lengths, and replicated key findings in independent cohorts of embryos using this second

methodology. Importantly, both assays verified the expected telomere elongation during preimplantation embryogenesis and also identified comparable relative changes between experimental models of mitochondrial stress and stress-reversing treatments.

Because our experiments measured telomere length at multiple points during preimplantation development, they revealed that distinctly different regulatory mechanisms control the first phase of

**Fig. 7 | Advanced maternal age impairs embryo telomere elongation that is restored by preconception treatment with mitochondria-acting therapeutics.** Female mice that were reproductively aged (12 months old) and young (3-4 months) controls were treated with BGP-15 (100 mg/kg) or saline vehicle before ovulation, IVF and embryo assessments (**a**) Created in BioRender. Gordon, Y. (2025) https://BioRender.com/c74d013. Zygotes were labelled with MitoSox Red (red, and Hoechst-3342 (blue)); and red fluorescence measured (**b**; $n = 21$ young, $n = 12$ aged, $n = 10$ aged+BGP-15). Morulae were stained with TMRM (red) and Hoechst-3342 (blue) and red fluorescence measured (**c**; $n = 3$ young, $n = 20$ young+BGP-15, $n = 14$ aged, $n = 8$ aged+BGP-15). Telomere length per cell (telomere/ $Rn18s$ ($2^{-\Delta\Delta Ct}$)) in individual MII oocytes of young vs 12-month-old mice (**d**; $n = 108$ young, $n = 66$ aged); as well as oocytes from 12 month old mice that were nulliparous vs multiparous (**e**; $n = 46$ nulliparous, $n = 37$ multiparous). Telomere length per cell in individual 8-cell embryos (**f**; $n = 36$ young, $n = 33$ young+BGP-15, $n = 19$ aged, $n = 44$ aged+BGP-15), whole blastocysts (**g**; $n = 31$ young, $n = 10$ aged, $n = 24$ aged+BGP-15)

and isolated ICM (**h**; $n = 12$ young, $n = 19$ aged, $n = 20$ aged+BGP-15). Metformin (2 mg/mL) or MitoQ (150 μM) was administered in drinking water to reproductively aged females for two weeks prior to gonadotropin stimulation and IVF of ovulated oocytes (**i**) Created in BioRender. Gordon, Y. (2025) https://BioRender.com/m49j079. Zygotes were labelled with MitoSox Red (red, and Hoechst-3342 (blue)); and red fluorescence measured (**j**; $n = 20$ young, $n = 31$ aged, $n = 21$ aged+Metformin, $n = 20$ aged+MitoQ). Telomere length in individual ICMs (**k**; $n = 47$ young, $n = 19$ aged, $n = 25$ aged+Metformin, $n = 16$ aged+MitoQ). Individual data points are plotted, horizontal lines are mean ± SEM (b, c, j). Violin plots show population distribution, and horizontal lines are mean ± SEM. qPCR data was log transformed for statistical analysis. Statistical tests were one-way ANOVA (b, c, f-h, j, k) or two-sided unpaired $t$-test (d, e); *$p < 0.05$, **$p \leq 0.0095$, ****$p < 0.0001$. For exact $P$ values see Supplementary Table 7. Representative images shown, scale bar: 20 μm (b, c, j). Source data are provided as a Source Data File.

telomere elongation compared to the second phase. Specifically, in each of the models of mitochondrial disruption (20% $O_2$ culture, rotenone, aging, obesity) telomere lengths were identical at the 8-cell stage, even in contexts where telomeres started out shorter in ovulated oocytes, i.e. with rotenone-exposure, aging or obesity. This indicates that the earliest phase of telomere elongation, which is thought to be mediated via the recombination-based Alternative Lengthening of Telomeres (ALT) mechanism[12,13], is not affected by the defects that are present in these oocytes. In contrast, each of the models of mitochondrial dysfunction exhibited the exact same defect in telomere elongation between the 8-cell stage and the ICM of the blastocyst. This phase of telomere elongation is known to be regulated by telomerase-mediated mechanisms[12–14], and suggests that mitochondrial dysfunction, and specifically increased mtROS, interferes with the normal functioning of this pathway.

The exact nature of the mitochondrial-nuclear communication that regulates preimplantation telomere elongation remains to be identified and represents a new area of investigation for the field. Our results indicate that the molecular machinery is already present within the ovulated egg. Specifically, oocytes from either lean or obese mice underwent fertilization and culture to blastocyst under identical conditions, and then embryos from both groups were transferred to standard surrogate mice for gestation; yet differences in telomere lengths in fetal tissues were observed. Further, this molecular communication can be modulated at the time of fertilization. When oocytes from obese (or rotenone-fed) mice were treated in vitro with BGP-15 (or MitoQ or SS-31) during IVF and embryo culture, deficiencies in ICM telomere length were mitigated. Most remarkably, pronuclear transfer experiments showed that the nuclear signal was established prior to the first embryonic cleavage. It is likely that zygotic mtROS levels are a key mediator, or a surrogate indicator, because their inverse correlation with ICM telomere length was a highly consistent phenotype across all of the different models of mitochondrial dysfunction and pharmacological manipulation, more so than changes in mitochondrial metabolites or membrane potential. The identity of the 'epigenetic' nuclear factor(s) involved also remain to be determined. ROS-induced 8-oxodG lesions are known to inhibit telomerase activity[54] and we show that increased 8-oxodG in zygotes correlates with shorter ICM telomeres in some contexts. In parallel, our immunostaining for 5mC/5hmC provided a cursory view of global epigenetic status, and specific methylation and acetylation patterns on telomeric chromatin are known to influence length homeostasis and acquisition of pluripotency[52,55]. Any number of other DNA and/or histone modifications could also influence telomere elongation directly via epigenetic modifications or indirectly via regulation of gene expression, for instance of telomerase subunits or cofactors. Additional work is required to identify specific regions or motifs that may be responsible in each of the different contexts.

These findings provide further evidence that maternal signals are transmitted to the next generation during the peri-conception period; and extend our understanding of genomic modifications that direct an offspring's lifetime health trajectory. Specifically, during the hours between ovulation and syngamy, a key developmental window, zygotic mitochondria respond acutely to maternal physiological or in vitro exogenous cues, signal to the embryonic pronuclei and 'program' the capacity for telomere elongation during the transition to pluripotency between the 8-cell and ICM formation. The demonstrated plasticity of oocyte mitochondrial responses, and our identification of specific pharmaceutical compounds that can modulate them during preconception and immediately following fertilization, means that there are therapeutic opportunities to optimize this biology which is a major determinant of chronic disease risk.

## Methods

### Animals, hormone treatment, and drug administration

All animal experiments were approved by the University of Adelaide's Animal Ethics Committee (Approval Nos: M-2017-116, M-2018-121, M-2018-086, M-2021-076, M-2021-096) and conducted in accordance with the Australian Code of Practice for the Care and Use of Animals for Scientific Purposes. Female mice: CBA/CaHArc, C57/BL6Arc, or CBA.F1 (CBA/CaHArc x C57/BL6Arc) at 6-7 weeks of age, and males at 6-8 weeks of age were obtained from the University of Adelaide's Laboratory Animal Services (CBA/CaHArc, CBA.F1) or Animal Resources Centre (C57BL6/Arc) (Western Australia). All mice were maintained in 12 h/12 h light/dark conditions with ambient temperature of 22°C and 40-60% humidity. Mice were given water and rodent chow (Teklad Global 19% Protein Extruded Rodent Diet, #2019) *ad libitum*. For all drug treatments or dietary manipulations, mice were randomly assigned to experimental groups.

Obese or reproductively aged female mice, as well as lean young controls were generated from the same colony (C57BL/6JSfdAnu-Alms1bbb/Apb mouse strain maintained as heterozygous breeding pairs), termed "Blobby" mice, and fed an identical standard chow diet as in refs. 33,34. Obese mice were homozygous (bbb/bbb) for the 'Blobby' mutation of the Alms1 gene which results in hyperphagia and profound obesity even when maintained on a standard mouse chow diet (Teklad Global Soy Protein-free Extruded Rodent Diet, #2020X)[34]. Females were deemed obese when they weighed at least 36 g, which occurred at 4-5 months of age, and wild-type littermates (i.e. aged matched) were used in parallel as lean controls. Reproductively aged females were wild-type or heterozygous (+/+ or bbb/+) at 12 months of age and young females (3-4 months old) were used in parallel as young controls.

Rotenone (#R8875) was sourced from Sigma and rotenone diet was prepared and used as in ref. 29. Specifically, control diet was a modified diet of AIN-93G (TD.97184), and Rotenone Diet was formulated at 150 ppm rotenone in TD.97184 meal diet by Teklad/Envigo exactly as for[29]. CBA.F1 mice at 6 weeks of age were randomly assigned

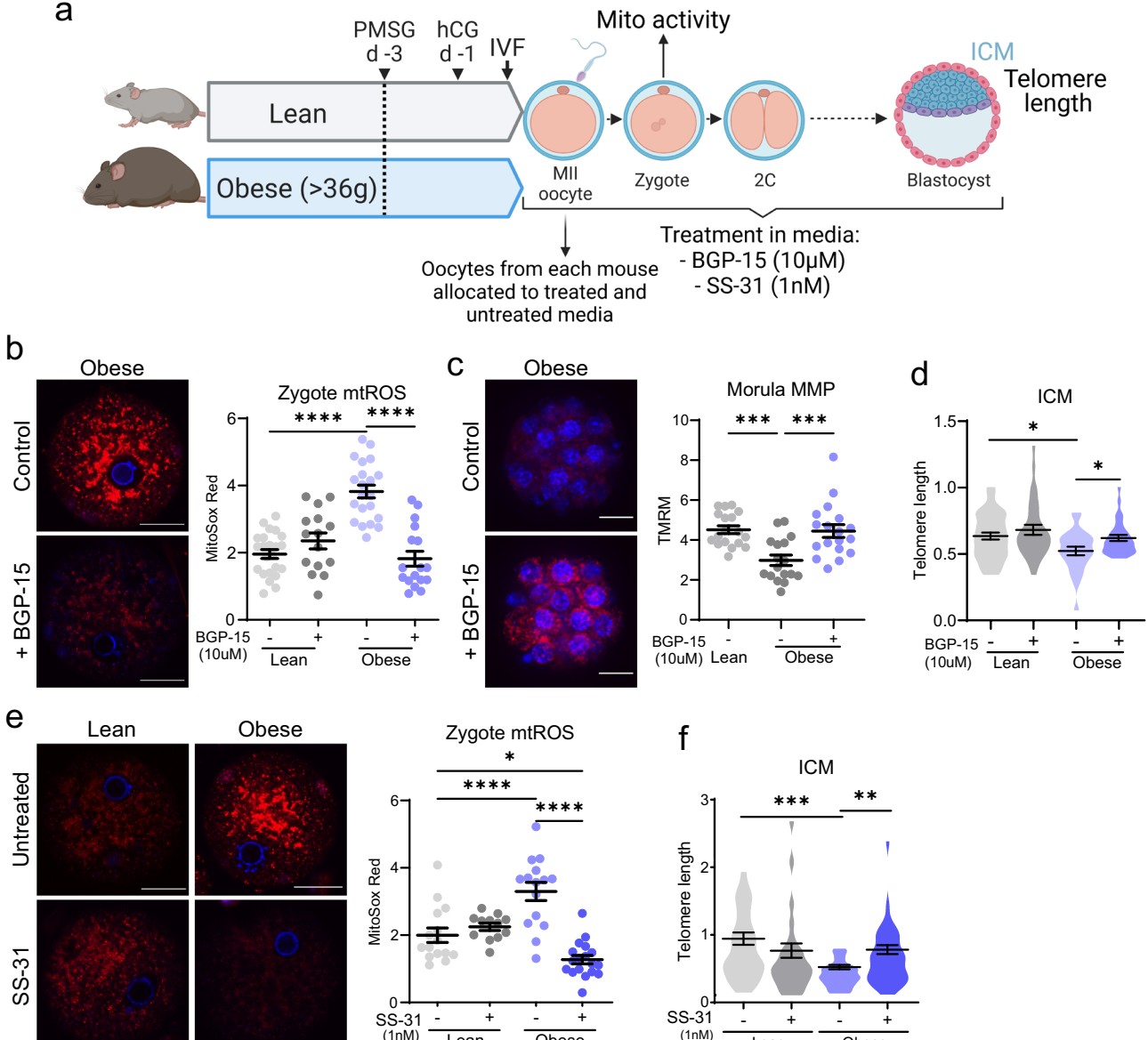

**Fig. 8 | Maternal obesity impairs embryo telomere elongation that is restored by treatment with mitochondria-acting therapeutics in vitro at fertilization.** Ovulated oocytes from lean or obese mice ( > 36 g) were cultured in media containing 10 µM BGP-15 or 1 nM SS-31 (or equal volume of vehicle (-)) from fertilization (**a**) Created in BioRender. Gordon, Y. (2025) https://BioRender.com/w92h854. Embryos treated with 10 µM BGP-15 were analyzed for MitoSox Red (red) and Hoechst-3342 (blue) in zygotes (**b**; $n = 23$ lean, $n = 15$ lean +BGP-15, $n = 21$ obese, $n = 17$ obese +BGP-15), TMRM (red) and Hoechst-3342 (blue) in morulae (**c**; $n = 18$ lean, $n = 17$ obese, $n = 18$ obese +BGP-15), and telomere length (telomere/ $Rn18s$ ($2^{-\Delta\Delta Ct}$)) in ICM (**d**; $n = 43$ lean, $n = 32$ lean +BGP-15, $n = 24$ obese, $n = 30$ obese +BGP-15).

Embryos treated with 1 nM SS-31 were analyzed for MitoSox Red (red) and Hoechst-3342 (blue) in zygotes (**e**; $n = 15$ lean, $n = 12$ lean +SS-31, $n = 15$ obese, $n = 18$ obese +SS-31) and telomere length (telomere/ $Rn18s$ ($2^{-\Delta\Delta Ct}$)) in ICM (**f**; $n = 29$ lean, $n = 29$ lean +SS-31, $n = 31$ obese, $n = 40$ obese +SS-31). Individual data points are plotted, horizontal lines are mean ± SEM (b, c, e). Violin plots show population distribution, and horizontal lines are mean ± SEM (d, Ff). qPCR data was log transformed for statistical analysis. Data analyzed using one-way ANOVA; *$p < 0.05$, **$p = 0.0013$, ***$p \le 0.0009$, ****$p < 0.0001$. For exact $P$ values see Supplementary Table 8. Source data are provided as a Source Data File.

to control or rotenone diet for three weeks as per our previous experiments[33]. This dose of rotenone is demonstrated to decrease Complex I- and Complex II-driven respiration without altering oxidative phosphorylation subunit abundance or causing overt physiological effects[29].

BGP-15 ([(O-[3-piperidino-2-hydroxy-1-propyl]-nicotinic amidoxime)], CAS 66611-37-8; Hangzhou Molcore Biopharmatech Co Ltd) was injected intra-peritoneally (i.p.) at 100 mg/kg of bodyweight in saline for 4 days, starting the day before PMSG injection (as in ref. 33,34). Controls were injected with 0.9% saline vehicle at the equivalent volume for body weight.

Metformin (1,1-Dimethylbiguanide hydrochloride, D150959, Sigma-Aldrich) was provided to mice at 2 mg/mL in drinking water *ad libitum* for 2 weeks, before PMSG and hCG treatment (as in ref. 33).

MitoQ (Mitoquinone Mesylate: [10-(4,5-Dimethoxy-2-methyl-3,6-dioxo-1,4-cyclohexadien-1-yl)decyl](triphenyl)phosphonium methanesulfonate) is a ubiquinone moiety linked to a lipophilic triphenylphosphonium cation by a 10-carbon alkyl chain that preferentially accumulates in mitochondria[36]. MitoQ (generously donated by MitoQ Ltd, New Zealand) was delivered at 150 µM in drinking water *ad libitum* for 2 weeks, before PMSG and hCG treatment (as in ref. 33).

Female mice were given pregnant mare serum gonadotropin (PMSG, #493-10, Lee BioSolutions) at 5IU/12 g body weight, followed by an equivalent dose of human chorionic gonadotropin (hCG, Pregnyl) 47.5 h later, each via intraperitoneal (i.p.) injection.

## Derivation of oocytes, early embryos, and blastocysts

For Meiosis II (MII) oocyte collection, mice were culled via cervical dislocation 15 h post-hCG administration, and ovaries and oviducts collected and placed in pre-warmed (37 °C) αMEM-HEPES (Gibco, #12000-022) handling media, supplemented with 1% fetal calf serum (FCS). Ovulated cumulus oocyte complexes (COCs) were isolated by opening the oviducts using a 30 G needle. Cumulus cells were completely removed from MII oocytes via treatment with hyaluronidase (Seikagaku, #100741) for 5-10 minutes at 37 °C, followed by aspiration using a glass pipette pulled to an appropriate diameter.

Embryos were generated at precise stages of development via in vitro fertilization (IVF). Briefly, ovulated COC clusters were gently washed twice in pre-warmed (37 °C) fertilization media (Vitro Fertilization; Cook Australia, Brisbane, Australia), before being placed in a 100 μL fertilization drop containing the equivalent of 10 μL of capacitated sperm from a male of proven fertility (referred to as 'fertilization time' or culture day 1), before being returned to the incubator (37 °C, 5% $O_2$, 6% $CO_2$) for 4 h. Fertilized oocytes were then cleaned of all excess sperm and cumulus cells via gentle aspiration and only those with 2PB were deemed to have fertilized and were transferred to a culture dish containing cleave media (Vitro Cleave; Cook Australia, Brisbane, Australia, 10 embryos per 20 μL cleave media drop) and returned to the incubator until 24 h post-fertilization time. At this time (i.e. culture day 2), the number of embryos that successfully reached the 2-cell stage was recorded. For in vitro BGP-15 treatment of oocytes and embryos, BGP-15 stock at 0.5 M in sterile $H_2O$ was diluted to 10 μM in the respective media (fertilization media or cleave media). For in vitro SS-31 treatment of oocytes and embryos, SS-31 (kindly provided by Hazel H. Szeto) stock at 1 μM in sterile $H_2O$ was diluted to 1 nM in the respective media. For controls, the equivalent volume of sterile $H_2O$ was added to the media. For high oxygen (20% $O_2$) experiments, the fertilization incubation and subsequent culture in cleave media were conducted at 37 °C, 20% $O_2$, 6% $CO_2$.

The number of ovulated oocytes per mouse and embryo on-time development was monitored in every experiment to ensure collection of embryos at the precise stage. 4-cell embryos were collected at 39 h post-fertilization, 8-cell at 55 h (culture day 3), morula at 77 h (culture day 4), and blastocysts at 96 h post-fertilization time (culture day 5). In all assays, only morphologically normal oocytes, zygotes and embryos were included. Morulae collected for analysis were all of a similar cell number, and blastocysts were all at a similar developmental stage (late-expanded or early hatching) and morphologically normal.

For treatment of oocytes and zygotes from the fertilization to 2PN window, BGP-15 (10μM), SS-31 (1 nM), or MitoQ (at 50 nM) was present in the fertilization media for 4 hours until presumptive zygotes were cleaned via gentle aspiration and transferred to cleave media containing the aforementioned compounds until a total exposure time of 8 hours. After this, zygotes were washed 5x in untreated (control) cleave media and cultured in control cleave media for the remaining duration.

For in vivo embryos, female mice were housed (1:1) with male mice of proven fertility after hCG administration and then separated after 16 hours. To collect 2-cell embryos, females were culled via cervical dislocation 24 h later (40 h post-hCG) and 2-cell embryos dissected from the oviduct. To collect blastocysts, females were culled 93 h post-hCG and embryos flushed from the uterus, using a 1cc syringe with pre-warmed αMEM-HEPES + 1% FCS handling media. Telomere length in embryos derived via IVF were directly compared to those conceived by in vivo fertilization (i.e. mating) and found to be not different (Fig. S3b, c).

## Isolation of blastocyst cell populations

Collection of paired Inner Cell Mass (ICM) and Trophectoderm (TE) cell populations (Fig. 1c) was by manual dissection. The zona pellucida was removed using Acid Tyrode's Embryomax solution (Millipore, #MR-004D) and blastocysts transferred to individual 5 μL pre-warmed (37 °C) drops of αMEM handling media with 1% FCS in a 50 x 9 mm petri dish (Falcon, #351006). Blastocysts were manually separated into their two cell populations using the Eppendorf TransferMan 4r system (Eppendorf), and a standard holding pipette with a 15 μm inner diameter and 120 μm outer diameter (The Pipette Company, #LHC-ID15) and biopsy pipette with a 30 μm tip (OD) and 90° bevel (The Pipette Company, #LBC-OD30-BA90). Cell populations were separated and frozen individually per blastocyst, but paired for analysis.

Immunosurgery was used to isolate a purified ICM population (see Fig. 1d). Blastocysts had zona pellucidae removed via treatment with Acidic Tyrode's Embryomax solution (Millipore, #MR-004D), washed twice in cleave media and transferred to cleave media containing 20% heat-inactivated rabbit anti-mouse serum (Sigma, #M5774). Following a 1 h incubation (37 °C, 5% $O_2$, 6% $CO_2$), blastocysts were washed three times in cleave media and transferred to cleave media containing 20% guinea pig serum (Sigma, #G9774) and incubated for 15 min. Blastocysts were washed three times in cleave media and incubated for a further 30 min in this media. Lysed TE cells were removed by gently aspirating ICMs in a finely drawn glass pipette.

## Zygotic pronuclear transfer

Female mice that consumed control or rotenone-containing diet for three weeks were given PMSG and hCG (as above, to precisely control the timing of ovulation and thus fertilization) and housed (1:1) with an 8-week-old CBA.F1 male to allow mating. 18 h post-hCG (i.e. ~6 h post-fertilization) females were humanely killed and presumptive zygotes collected, fully denuded of cumulus cells, transferred to cleave media and placed in the incubator for 1.5-2 h (~8 h post-fertilization). In vivo fertilized zygotes were confirmed to have elevated ROS levels (Fig. S10a). After this, zygotes were incubated in cleave media containing 1 μg/mL Cytochalasin D (Sigma, #C2618) and 0.3 μg/mL Nocodazole (Sigma, #SML1665) for 20 minutes. Zygotes were then transferred to αMEM-HEPES handling media containing 60 μg/mL BSA (Sigma, #A9418), 1 mg/mL polyvinylpyrrolidone (PVP, Sigma, #PVP360), and Cytochalasin D and Nocodazole at the above concentrations. Both pronuclei were removed from a zygote by micropipette (Eppendorf, #5195000079) followed by a small volume of inactivated Sendai Virus (Cosmo Bio, USA; ~3000 hemagglutinating units/ml) being drawn into the pipette. The virus and pronuclei were injected into the peri-vitelline space of a second enucleated one-cell embryo (see Fig. S10b). Reconstructed embryos of each type were generated on the same day, in random order and within 2 h, to ensure identical developmental stages in all groups. Reconstructed embryos were washed in cleave media to remove inhibitors, and cultured in cleave media as above to the blastocyst stage. Blastocyst development rates were similar to non-reconstructed embryos and not different between groups (Fig. S10c–e).

## Telomere and Rn18s qPCR using DNA from individual oocytes and embryos

Individual oocytes or embryos for qPCR analysis were washed three times in 1x phosphate buffered saline (PBS) containing 1 mg/mL PVP, and transferred in 1 μL to a 0.5 mL PCR tube (Axygen, #PCR-05-C), snap frozen in liquid nitrogen ($LN_2$), and stored at -80 °C until use. DNA was extracted by adding 9 μL of lysis buffer (50 mM Tris-HCl pH 8.0, 1 mM EDTA, 200 μg/mL Proteinase K, 0.5% Tween-20) to each sample (final volume of 10 μL) and heating to 55 °C for 2 h, followed by 95 °C for 10 minutes. Samples were cooled to 4 °C and briefly centrifuged. ICM samples were further diluted with an additional 10 μL of sterile $H_2O$ to give a total volume of 20 μL.

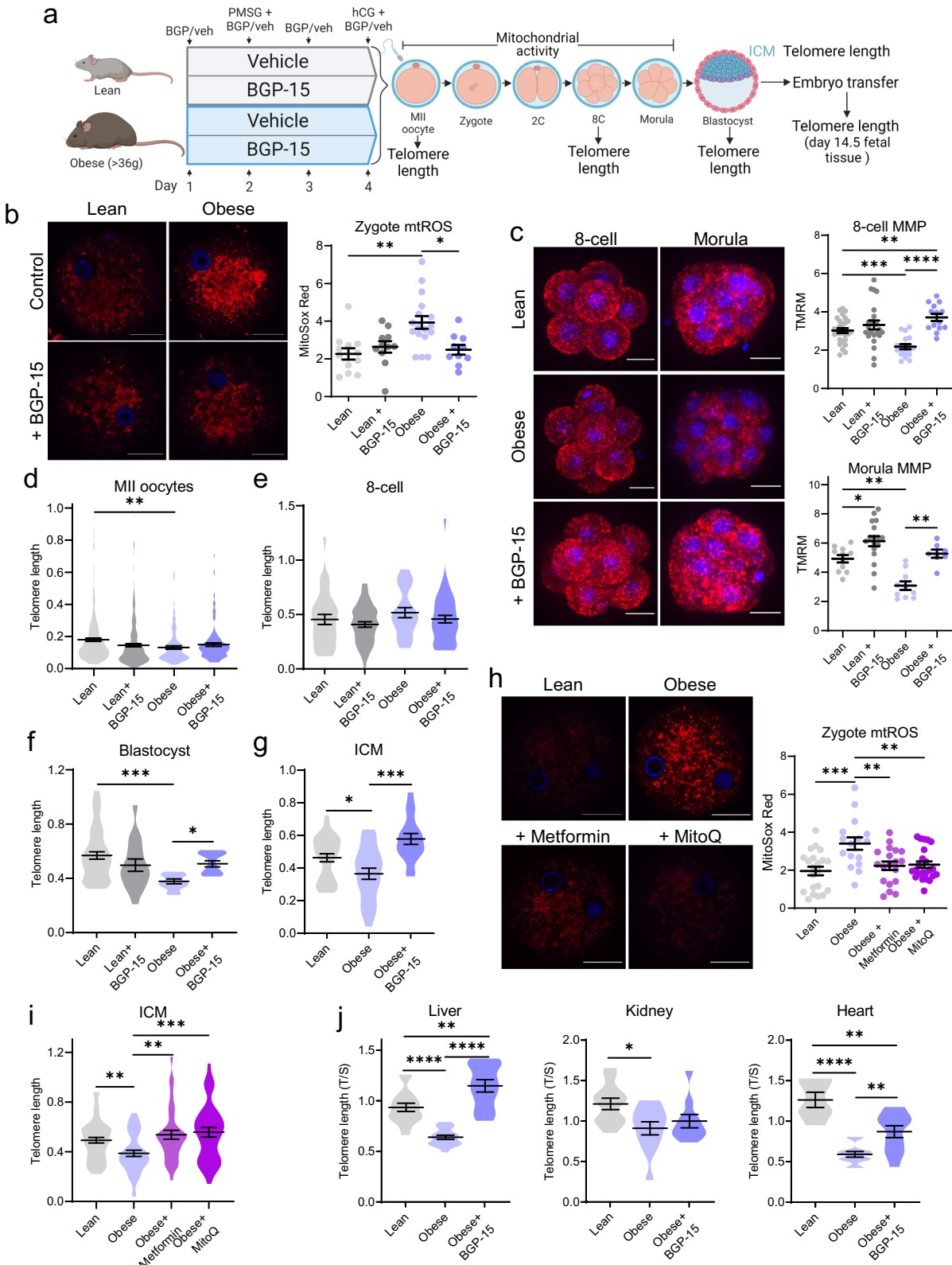

To generate 'calibrator' DNA for use as a standard within every assay, a whole ovary from a 6-week-old PMSG- and hCG-stimulated CBA female mouse was collected and total DNA extracted using the QIAmp DNA Micro Kit (Qiagen, #56304) 'Isolation of Genomic DNA from Tissues' protocol, according to manufacturer's instructions. DNA concentration and purity were quantified by NanoDrop One Micro-volume UV-Vis Spectrophotometer and diluted to 2.5 ng/µL.

qPCR was performed using Power SYBR Green PCR Master Mix (Applied Biosystems, #4367659), standard MicroAmp Optical 96-well Reaction Plates (Life Technologies, #4306737), and custom-made primers (Sigma), using an Applied Biosystems 7900HT Fast Real-Time PCR system (Applied Biosystems) or Quantstudio 12K Flex (Thermo-Fisher). Cycling conditions for all primers were set at 95 °C for 10 minutes, followed by 40 cycles of 95 °C for 15 s, 60 °C annealing for

**Fig. 9 | Maternal obesity impairs embryo and fetal telomere elongation that is restored by pre-conception treatment with mitochondria-acting therapeutics.** Obese female mice (>36 g) and lean littermate controls were treated with BGP-15 (100 mg/kg) or saline vehicle prior to ovulation, IVF and analysis of embryos and fetal tissues (**a**) Created in BioRender. Gordon, Y. (2025) https://BioRender.com/z87s922. Zygotes were labelled with MitoSox Red (red) and Hoechst-3342 (blue) and red fluorescence measured (**b**; $n = 12$ lean, $n = 11$ lean+BGP-15, $n = 17$ obese, n = 10 obese+BGP-15). 8-cell embryos and morulae were stained with TMRM (red) and Hoechst-3342 (blue) and red fluorescence measured (**c**; 8-cell: n = 28 lean, $n = 23$ lean+BGP-15, $n = 17$ obese, $n = 14$ obese+BGP-15; morula: $n = 11$ lean, $n = 17$ lean+BGP-15, $n = 10$ obese, $n = 7$ obese+BGP-15). Telomere length per cell (telomere/$Rn18s$ ($2^{-\Delta\Delta Ct}$)) was assessed by qPCR in individual MII oocytes (**d**; $n = 250$ lean, $n = 218$ lean+BGP-15, $n = 104$ obese, $n = 100$ obese+BGP-15), 8-cell embryos (**e**; $n = 30$ lean, $n = 50$ lean+BGP-15, $n = 22$ obese, $n = 45$ obese+BGP-15), whole blastocysts (**f**; $n = 51$ lean, n = 16 lean+BGP-15, n = 11 obese, n = 12 obese+BGP-15) and ICM (**g**; $n = 31$ lean, $n = 24$ obese, $n = 17$ obese+BGP-15). Metformin (2 mg/mL) or MitoQ (150 μM) was administered in drinking water to obese females for two weeks prior to gonadotropin stimulation and IVF of ovulated oocytes. Zygotes were labelled with MitoSox Red (red) and Hoechst-3342 (blue); and red fluorescence measured (**h**; $n = 21$ lean, $n = 16$ obese, $n = 19$ obese+Metformin, $n = 23$ obese+MitoQ). Telomere length in individual ICMs (**i**; $n = 57$ lean, $n = 45$ obese, $n = 36$ obese+Metformin, $n = 37$ obese+MitoQ). Blastocysts from oocytes of lean, obese, or obese BGP-15-treated (100 mg/kg) mice were transferred to surrogate females for gestation and fetal tissues collected at day 14.5 of pregnancy for qPCR telomere length analysis (telomere/$36b4$ ($2^{-\Delta\Delta Ct}$)) of liver, kidney, and heart (**j**; liver: $n = 16$ lean, $n = 17$ obese, $n = 12$ obese+BGP-15; kidney: $n = 12$ lean, $n = 12$ obese, $n = 10$ obese+BGP-15, heart: $n = 6$ lean, $n = 12$ obese, $n = 11$ obese+BGP-15). Individual data points are plotted, horizontal lines are mean ± SEM (b, c, h). Violin plots show population distribution, and horizontal lines are mean ± SEM (d-g, i, j). qPCR data was log transformed for statistical analysis. Data analyzed using one-way ANOVA; *$p < 0.05$, **$p < 0.009$, ***$p \leq 0.0007$, ****$p < 0.0001$. For exact $P$ values see Supplementary Table 9. Source data are provided as a Source Data File.

30 s and extension at 72 °C for 30 s, followed by standard melt curve cycling. Two replicate reactions were performed for each sample and each primer pair, using 2 μL of sample per reaction (see Fig. S1a). The final reaction for each well was 10 μL Power SYBR Green PCR Master Mix, 0.5 μL each of forward and reverse primer (stock at 10 μM), 2 μL of sample, and sterile $H_2O$ to a final volume of 20 μL. Telomere primer sequences were: telomere forward 5′-CGG TTT GTT TGG GTT TGG GTT TGG GTT TGG GTT TGG GTT-3′, telomere reverse 5′-GGC TTG CCT TAC CCT TAC CCT TAC CCT TAC CCT TAC CCT-3′[56]. Reference gene primer sequences were: $Rn18s$ forward 5′-AGA AAC GGC TAC CAC ATC CAA-3′, $Rn18s$ reverse 5′-CCT GTA TTG TTA TTT TTC GTC ACT ACC T-3′[57]. ($Rn18s$ was used as the reference gene for all oocyte/ embryo samples because it is a multi-copy gene; and thus detectable in samples with low nuclear DNA, i.e. individual oocytes or 2-cell embryos.) To minimize plate-to-plate variation, reactions for each sample were run on the same plate, and calibrator DNA included on every plate. Telomere relative to $Rn18s$ copies was calculated using the $2^{-\Delta\Delta Ct}$ method, where the ΔCt of the calibrator (telomere Ct− $Rn18s$ Ct) was subtracted from the ΔCt of the sample (telomere Ct − $Rn18s$ Ct) to give the ΔΔCt.

Embryo sex determination qPCR was performed using the same reagents and systems as above, with two reactions performed for every sample. The final reaction for each sample consisted of 10 μL Power SYBR Green PCR Master Mix, 0.5 μL each of forward and reverse primer ($Sry$ primers, Sigma, stock at 10 μM, forward sequence 5′-AAG CGC CCC ATG AAT GCA TT-3′, reverse sequence 5′-TCC CAG CTG CTT GCT GAT CT-3′), 4 μL of sample, and sterile $H_2O$ to a final volume of 20 μL. To confirm qPCR product size, reactions were run on a 4% agarose (Promega, #V3125t) gel with Gel Red (Biotium, #41003). Each sample had 2.5 μL of 6x loading dye (New England Bio Labs, #B70245) added and 10 μL run per lane using a 100 bp ladder (ThermoFisher Scientific, #SM0243) for reference. Gels were run at 100 V for 1 h and imaged via Biorad Gel Doc EZ Imager with Image Lab software. Male embryos were distinguished by a PCR product at 105 bp. Telomere length was not different between ICM from male and female embryos (see Fig. S3a), thus, unless specifically stated, male and female embryos were grouped for analysis.

## Chromosome spreads and telomere/centromere fluorescence in situ hybridization (FISH)

2-cell and 4-cell embryos were treated with 100 ng/ml Colcemid (Gibco, #15212012) overnight to enrich for mitotic cells before chromosome spreads. Oocytes or embryos were exposed to Acid Tyrodes solution for 1 min at 25 °C to remove the zona pellucida. After a 2-min recovery in M2 medium (Sigma, #M7167), samples were transferred into a 0.5% Sodium Citrate drop for 2 min and then moved onto a clean microscope slide that had been previously dipped in a solution of 1% paraformaldehyde in MilliQ water, pH9.2, containing 0.15% Triton X-100 (Sigma, #T8787) and 3 mM dithiothreitol (Sigma, #43816). The slides were allowed to dry slowly in a humid chamber overnight, and then dehydrated with 75%, 85% and 100% ethanol 5 min each. After dehydration, slides were dried for several minutes, then dipped into a canister with 70% formamide and 30% 2×SSC (saline citrate) solution at 80 °C for 5 min. Slides were dehydrated with iced 75%, 85% and 100% ethanol and then dried again. Slides were incubated with telomere C-strand AlexaFluor 488-conjugated telomere PNA (peptide nucleic acid) probe (1:200, PNA Bio Inc.) diluted in 50% formamide containing 2% salmon sperm DNA (15632-011, Invitrogen) overnight at 37 °C in humid chamber. Slides were washed with 70% formamide containing 10 mM Tris-HCl (pH 7.5) then incubated with CENP-B Cy3-conjugatede centromere PNA probe (1:100, denatured at 80 °C for 3 min, PNA Bio Inc.) diluted in 30% formamide containing 2% salmon sperm DNA (15632-011, Invitrogen) overnight at room temperature. Slides were sequentially washed with 4×SSC, 2×SSC, 1×SSC, and 0.5×SSC at 50 °C for 5 min each. Finally, slides were dehydrated with 75%, 85% and 100% ethanol for 5 min each, and DNA was stained with Hoechst 33342 (10 μg/ml) for 10 min. All images were collected using a Zeiss Imager M2 fluorescence microscope linked to an AxioCam MRm CCD camera system and processed using the Zen software 2011 (Carl Zeiss Microscopy). To examine the relative abundance of telomere and centromere FISH signals, a threshold level for each channel was set independently using ImageJ to measure the mean of fluorescence intensity above the threshold level.

## Inner Cell Mass telomere fluorescence in situ hybridization (FISH)

Purified ICMs (via immunosurgery) were incubated in accutase (Gibco, #A11105-01) diluted 6:4 in PBS-PVP under paraffin oil at 37 °C for up to 60 min. ICMs were gently pipetted using a finely drawn glass pipette every 5–10 min to assist with dissociation until fully dissociated. Cells were transferred to a slide in a minimal volume to an area (approximately 3 mm x 6 mm) marked using a PAP pen (Sigma). Hypotonic solution (0.5% sodium citrate) was added to cover the defined area and slides were incubated for 10 min in a humidified chamber (to prevent evaporation). Following this, as much hypotonic solution as possible was removed and fixation solution (3:1 methanol: acetic acid) was added to cover the defined area and allowed to air dry.

Slides were treated with 100 μg/mL DNase-free RNase A (Sigma) in 2× SSC for 30 min at 37 °C, rinsed in PBS, and postfixed in 4% (vol/vol) formaldehyde in PBS at room temperature for 10 min. Slides were then dehydrated in a graded ethanol series (75% (v/v) for 3 min, 85% (v/v) for 2 min and 100% for 2 min) and allowed to air dry. Dehydrated slides were then overlaid with 0.3 μg/ml Alexa 488−OO-(TTAGGG)$_3$ telomeric probe (HLB Panagene) in PNA hybridization solution (70% deionized formamide, 0.1 μg/ml herring sperm DNA, 10 mM Tris−HCl, pH 7.5, 4 mM $Na_2HPO_4$, 0.5 mM citric acid, and 1.25 mM $MgCl_2$), denatured at 80 °C for 3 min, and hybridized at room temperature overnight. Slides were washed in PNA wash A (70% formamide, 10 mM Tris pH 7.5) and

PNA wash B (50 mM Tris pH 7.5, 150 mM NaCl, 0.8% Tween-20) for 15 min each, rinsed briefly in deionized water, air dried and mounted in ProLong™ Gold Antifade with DAPI (Thermofisher Scientific, #P36931). Microscopy images were acquired on a Zeiss Axio Imager microscope with appropriate filter sets.

Automated image analysis: ZEN microscopy images (.czi) were processed into extended projections of z-stacks using ZEN 3.4 software (Zeiss) and imported into Cellprofiler v4.2.1 for analysis. The DAPI channel was used to mask individual nuclei as primary objects. Foci within each segmented nucleus were identified using an intensity-threshold based mask.

Standards for this assay (see Fig. S2f) are cell lines known to have homogeneous short telomere lengths (HCT116, HeLA) or long heterogeneous telomeres (U-2 OS). HCT116, HeLa, and U-2 OS cancer cell lines were sourced in house or from CellBank Australia, tested clear for Mycoplasma, and were verified by STR profiling through CellBank Australia.

## MitoSOX Red (MSR) staining
Zygotes were collected and incubated for 20 minutes at 37 °C in handling media containing 60 µg/mL BSA, 1 mg/mL PVP, 5 µM MitoSOX Red (Invitrogen, #M36008) with Hoescht-3342 (1 µg/mL, Sigma, #H33342). Samples were washed briefly in handling media before being mounted in pre-warmed handling media (containing BSA and PVP, as above) and imaged using an Olympus FV3000 Confocal Microscope at 60x magnification, with a single z-plane chosen for each zygote where the diameter was largest. Images were acquired using the same confocal microscope settings with the operator blinded to treatment group. Total cell fluorescence was calculated and adjusted for background fluorescence using Image J to give the corrected total cell fluorescence (CTCF).

## NAD(P)H and FAD++ imaging
Fertilized zygotes were collected and imaged using handling media containing 60 µg/mL BSA, 1 mg/mL PVP at 37 °C. The reduced nucleotides NAD and NADPH (NAD(P)H) were excited by ultraviolet light (405 nm), and autofluorescence collected using a 435–485 nm emission detection bandwidth. FAD++ autofluorescence was excited using a 488 nm laser and collected with a 505-550 nm emission detection bandwidth. Signals were captured using an Olympus FV3000 Confocal microscope at 60x magnification, with a single z-plane chosen for each zygote where the diameter was largest. Total cell fluorescence was calculated and adjusted for background fluorescence using Image J to give the corrected total cell fluorescence (CTCF).

## MitoTracker green staining
Fertilized zygotes were labelled with MitoTracker Green (100 nM, Invitrogen, #M7514) and Hoescht-3342 (1 µg/mL) for 20 min at 37 °C in handling media also containing 60 µg/mL BSA, 1 mg/mL PVP. Samples were washed briefly in handling media before being mounted in pre-warmed handling media (as above) and imaged using a Cell Voyager CV1000 spinning disc confocal (Yokogawa) with a 40x objective. Zygotes were imaged using a z-stack with images captured 0.5 µm apart. All images were acquired using identical settings with the operator blinded to treatment group. Z-stack images were compressed using the "Z-project" function on Image J and total cell fluorescence was determined and adjusted for background fluorescence to give the corrected total cell fluorescence (CTCF).

## Mitochondrial membrane potential assay
Fully denuded MII oocytes or fertilized zygotes, or embryos at the indicated developmental stage were washed twice in pre-warmed (37 °C) handling media containing 60 µg/mL BSA, 1 mg/mL PVP and incubated for 30 min at 37 °C in handling media containing BSA and PVP, as above, and 25 nM TMRM (Tetramethylrhodamine methyl ester perchlorate; Sigma, #T5428) with 1 µg/mL Hoescht-3342 added for the final 15 min. As validation of the assay specificity, a cohort of zygotes was then exposed to 1 µM FCCP (Carbonyl cyanide-p-tri-fluoromethoxyphenylhydrazone, Sigma, #C2920) for 10 minutes before being imaged as an MMP uncoupling control (Fig. S5g). Samples were washed briefly in handling media before being mounted in pre-warmed handling media (as above) and imaged using a Cell Voyager CV1000 spinning disc confocal (Yokogawa) with a 40x objective. Whole embryos were imaged using a z-stack with images being captured 0.5 µM apart. Images were acquired using the same confocal microscope settings with the operator blinded to treatment group. Z-stack images were compressed using the "Z-project" function on Image J and total cell fluorescence was calculated and adjusted for background fluorescence to give the corrected total cell fluorescence (CTCF).

## 8-oxodG immunofluorescence
8-oxodG in zygotes was measured by immunohistochemistry and expressed as total fluorescence in pronuclei. Zygotes were collected 8 h after fertilization and the zona pellucida dissolved using Acid Tyrode's Embryomax solution (Millipore, #MR-004D), followed by a brief wash in phosphate buffered saline (PBS) containing 1 mg/ml PVP (PBS-PVP). Zygotes were fixed in -20 °C methanol (overnight) before being washed in PBS-PVP twice. Zygotes were permeabilized in 0.2% Triton X-100 for 10 min at room temperature (RT), followed by 2x 10 min washes in PBS-PVP. Zygotes were treated with RNase A (5 mg/mL, Sigma, #51657223) for 1 h at 37 °C and washed 3x 10 min in PBS-PVP. Zygotes were incubated in 4 N HCl solution (Honeywell, Fluka #84435) for 10 minutes at RT, and washed thoroughly in 0.05% Tween-20 in PBS (PBS-T). Zygotes were blocked overnight at 4 °C in BlockAce (Bio-Rad, #BUF029, made according to manufacturer's instructions), followed by overnight incubation at 4 °C in mouse anti-8-oxoG (Nikken SEIL Co. Ltd, #MOG-100P, clone N45.1, Lot. 013 MOG-100P) diluted 1:10 in BlockAce (made according to manufacturer's instructions, then diluted 1:10 in PBS for use). Zygotes were washed 3x for 10 min in PBS-PVP containing 0.1% Triton X-100 (BPD), and incubated in Alexa Fluor 594-conjugated goat anti-mouse (Invitrogen, #A11032) at 1:300 in BlockAce (diluted as above for primary antibody) for 1 h at RT, followed by another 3x 10 min washes in PBD. Zygotes underwent a final 10 min wash in PBS-PVP before being mounted on slides with ProlongTM Diamond Antifade Mountant (Invitrogen, #P36965). Zygotes were imaged using a z-stack and captured 0.5 µm apart, using the same confocal microscope settings with the operator blinded to treatment group. Z-stack images were compressed using the "Z-project" function on Image J and total fluorescence in both pronuclei was calculated and adjusted for background fluorescence to give the corrected total cell fluorescence (CTCF).

## Pronuclear cytosine methylation immunofluorescence
Nuclear 5mC and 5hmC in zygotes was measured by immunohistochemistry and expressed as total fluorescence and as a differential between the male and female pronuclei, as in ref. 58. Briefly, fertilized zygotes (stage PN3 to PN4) were collected 10 h after fertilization. The zona pellucida was dissolved using Acid Tyrode's Embryomax solution (Millipore, #MR-004D), followed by a brief wash in phosphate buffered saline (PBS) containing 1 mg/ml PVP (PBS-PVP). Zygotes were fixed in 3.7% PFA-PBS for 20 minutes at room temperature (RT), followed by two five-minute washes in PBS-PVP, then permeabilized in 0.2% Triton X-100 (in PBS) for 10 min at room temperature, followed by three five-minute washes in PBS-PVP. Zygotes were incubated in 4 N HCl solution for 10 min at RT and washed thoroughly in 0.05% Tween-20 in PBS (PBS-T). Zygotes were blocked overnight at 4 °C in blocking solution (1% BSA, 0.02% Triton X-100 in PBS), then incubated overnight at 4 °C with mouse anti-5mC (1:200, Biorad, #MCA2201, clone 33D3, Lot. 170919), and rabbit anti-5hmC (1:600, Active Motif, #39769, Lot.

01218002), diluted in blocking solution. Following three ten-minute washes in PBS-T, samples were incubated for one hour at RT, while protected from light, with Alexa Fluor 594-conjugated goat anti-mouse (1:500, Invitrogen, #A11032) and Alexa Fluor 488-conjugated goat anti-rabbit secondary antibodies (1:1000, Invitrogen, #A11034). Experiments included negative controls in which the primary antibody was omitted (Fig. S6e). Samples were mounted on slides with ProlongTM Diamond Antifade Mountant (Invitrogen, #P36965) and examined using a Cell Voyager CV1000 spinning disc confocal (Yokogawa) with a 40x objective. Images were acquired using the same confocal microscope settings with the operator blinded to treatment group. Pronuclear size and staining intensity was calculated using ImageJ software. Paternal pronuclei are definitively identified by their larger size, which also validates identical developmental stages between groups. Mean signal intensity was calculated first by subtracting the signal from the cytoplasmic area (representing background staining), then dividing by the area of the pronuclei. The signal intensity of each marker was also expressed as a ratio of the values within the paternal versus maternal pronucleus of the same zygote.

### Transfer of blastocyst embryos to pseudo-pregnant female mice

Embryos from high oxygen culture or in vivo rotenone experiments were vitrified at the morula stage to later be thawed, cultured to blastocyst and transferred to pseudopregnant females. To determine whether embryo vitrification impacts ICM telomere length, vitrified morulae were thawed and cultured overnight prior to ICM collection and analysis. ICM telomere length was not different between those derived from fresh or vitrified morulae, regardless of oxygen culture conditions (Fig. S4a) or pre-conception rotenone exposure (Fig. S9a).

Female mice confirmed to have mated with a sterile vasectomized male were considered day 0.5 of pregnancy and selected as embryo recipients. At day 2.5 of pseudopregnancy, 12-16 fresh (i.e. continuously cultured) blastocysts were transferred to uteri (6–8 blastocysts/horn) of each pseudopregnant mouse anesthetized via isoflurane gas. Buprenorphine analgesia was injected subcutaneously once after the surgery. Mice were humanely killed by cervical dislocation on day 18.5 of pregnancy. Implantation sites were counted and outcomes (i.e. fetus or resorption) documented. Fetal weight, crown to rump length, placenta weight, and fetal sex were recorded. Fetuses were humanely euthanized and multiple tissues collected (heart, liver, kidney, tail, gonads, brain) and individually snap frozen in liquid $N_2$ and stored at -80 °C until use. Embryos cultured at 20% $O_2$ exhibited reduced implantation potential compared to those cultured at 5% $O_2$ (Fig. S4) and thus fetal tissues from the two groups could not be directly compared and were not examined further. Embryos from mice fed rotenone exhibited identical implantation rates as those from control mice (Fig. S9).

Experiments involving the obese mouse model used embryos generated by IVF and cultured at atmospheric oxygen to the blastocyst stage followed immediately by uterine transfer to surrogate females[34]. Female mice confirmed to have mated with a sterile vasectomized male were considered day 0.5 of pregnancy and selected as embryo recipients. At day 2.5 of pseudopregnancy, 7–10 fresh (i.e. continuously cultured) blastocysts were transferred to uteri (3–5 blastocysts/horn) of each pseudopregnant mouse anesthetized by i.p. injection of Avertin (0.5 mg/g body weight, Sigma-Aldrich, St. Louis, MO). Analgesia Carprofen (5 mg/kg) (Rimadyl Pfizer) was injected subcutaneously once after the surgery. Fetal tissues were collected at day 14.5 of pregnancy, and fetal tissue was extracted[34].

### In vivo-conceived fetuses

To generate naturally conceived (in vivo) fetuses, females that had been on rotenone or control diet for 3 weeks were housed (1:1) with 8-week old CBA.F1 males and presence of a copulatory plug was considered day 0.5 of pregnancy. Males were not exposed to the rotenone

diet (i.e. given control diet), and all females were placed onto control diet after presence of a copulatory plug to restrict rotenone exposure to the pre-conception period. Fetuses were collected on day 18.5 of gestation, and fetal number, weight, and sex ratios were not different (Fig. S7), and fetal tissues were collected.

### Fetal tissue DNA extraction and qPCR

Fetal tissues were lysed overnight at 55 °C with constant shaking in 250 μL of lysis buffer (20 mM EDTA pH 8.0, 50 mM Tris pH 8.0, 120 mM NaCl, 1% SDS) with 5 μL of Proteinase K (10 mg/mL, Sigma, #1.24568). The following morning, 250 μL of 4 M ammonium acetate was added, briefly vortexed, then incubated for 15 minutes at RT with shaking, then a further 10 min without. Samples were centrifuged at 9,400 x $g$ for 10 min and 400 μL of supernatant removed and added to a clean tube. 800 μL of 100% ethanol was added and briefly vortexed prior to centrifugation at 9,400 g x $g$ for 8 min to pellet the DNA. Ethanol was removed and the pellet was washed using 500 μL of 70% ethanol. Excess ethanol was removed and DNA was re-suspended by incubation at 55 °C with gentle shaking in 300 μL of sterile $H_2O$. DNA was quantified using a ThermoFisher Nanodrop One UV-Vis spectrophotometer (ThermoFisher) and diluted to 4 ng/μL with sterile water for use.

Telomere length was analyzed in fetal tissues via qPCR as described above using a Quantstudio 12 K Flex (ThermoFisher). Telomere and reference gene (*Rn18s*) primer sequences were as above. Fetal tissues from obese (and lean control) mice were analyzed using the reference gene *36b4*. Primer sequences (Sigma) were: *36b4* Forward: 5' ACT GGT CTA GGA CCC GAG AAG 3'; and *36b4* Reverse: 5' TCA ATG GTG CCT CTG GAG ATT 3' as in ref. 56. Cycling conditions were as above. The final reaction for each well was 10 μL Power SYBR Green PCR Master Mix, 0.2 μL each of forward and reverse primer (stock at 10 μM), 2 μL of sample, and sterile $H_2O$ to a final volume of 20 μL, and two replicate reactions were performed for each sample. Telomere relative to reference gene copies (T/S ratio) was calculated using the formula 1/(telomere Ct/ reference Ct) as in ref. 56.

### Statistical analysis

Results are presented as mean ± SEM. All data points represent independent biological replicates. Telomere length data is displayed as mean ± SEM overlaid on a violin plot to illustrate the distribution of data points, with the n-values indicated in the legend. Analysis of normality showed that telomere data required log transformation for statistical analysis. Statistical analysis was performed using Graph Pad Prism version 10.1.0 for Windows (GraphPad Software Inc., La Jolla, CA) and SPSS Statistics 29.0.1.0 (IBM, Armonk, NY). Paired two-tailed *t*-test, unpaired two-tailed *t*-test, one-way and two-way analysis of variance (ANOVA, comparison of means), and linear-mixed effects models were used as indicated and statistical significance was considered at *P*-value < 0.05. Where statistical significance is indicated with an asterisk, the *P*-value is provided in the legend.

### Reporting summary

Further information on research design is available in the Nature Portfolio Reporting Summary linked to this article.

## Data availability

All data that support this study are available in the main text or the supplementary materials. Source data are provided with this paper.

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

## Acknowledgements

This work was funded by the National Health and Medical Research Council APP1130364 (R.L.R., H.A.P., D.L.R., J.C.); a Channel 7 Children's Research Foundation Research Grant 20669397 (Y.E.W.); a National Health and Medical Research Council Research Fellowship APP1117975 (R.L.R.); a National Health and Medical Research Council Investigator Grant APP1194141 (M.A.F.) and an ARC Discovery Project Grant DP240101869 (H.A.P.) The authors acknowledge the technical assistance and expert advice of Dr Macarena Gonzalez and Haley Connaughton. Prof Jose Polo and Dr Carmen Williams contributed constructive comments on the manuscript. Biorender was used to create the schematics.

## Author contributions

Conceptualization: R.L.R. Methodology: Y.E.W., R.D.R., A.P.S., L.L.W., D.A., J.C., H.A.P., D.L.R. Investigation: Y.E.W., R.D.R., A.P.S., L.L.W., Q.-H.Z., D.A., J.K.W., S.G.P. Resources: L.H.W., H.H.S., P.Q.T., M.A.F., J.C. Visualization: Y.E.W., R.L.R. Supervision: J.C., H.A.P., D.L.R., R.L.R. Acquisition of Funding: R.L.R., H.A.P., D.L.R., J.C., Y.E.W. Writing—original draft: Y.E.W. Writing—review & editing: Y.E.W., M.A.F., J.C., D.L.R., R.L.R.

## Competing interests

Y.E.W. and R.L.R. are inventors on a patent application (PCT/US23/69908) related to this work. R.L.R. is a consultant with Mitochon Technologies and Vitaleon Pharma. M.A.F. is a shareholder and consultant with N-Gene Research Laboratories Inc and is founder and shareholder of Celesta Therapeutics. H.H.S. is the inventor of SS-31 (USPTO 7,576,601; August 18, 2009). The remaining authors declare no competing interests.
