## [Transparent Peer Review file · Nature Communications]

Telomere length in offspring is determined by mitochondrial-nuclear communication at fertilization

Corresponding Author: Professor Rebecca Robker

Version 0:

Reviewer comments:

Reviewer #1

(Remarks to the Author)

The work titled "Telomere length in offspring is determined by mitochondrial-nuclear communication at fertilization" studies the effects of oxidative stress 1) preconception (with rotenone given for 3 weeks prior to conception), 2) with IVF performed under optimal (5%) or high (20%) oxygen, 3) in aged (one year vs 3-4 months) or 4) obese female mice on embryonic (including ICM) telomere length, mitochondrial mass and membrane potential, global 5m and 5hm and offspring telomere length. The authors also added several molecules to counteract these effects, specifically BGP-15 (an antioxidant and PARP inhibitor), MitoQ and metformin.

In addition, pronuclear transfer experiments were performed to evaluate how the PN or mitochondria affect telomere length.

The authors found that:

1. mitochondrial-nuclear communication at fertilization controls telomere elongation during formation of the pluripotent Inner Cell Mass (ICM). In particular (fig 2B), PN transfer of rotenone exposed PN into control cytoplasm resulted in shorter ICM telomeres, while transfer of control PN into rotenone exposed cytoplasm did not shorten the telomeres, suggesting that the PN is the locus responsible to regulate telomere elongation
2. Oxidative stress reduces the telomere elongation that occurs at the morula and blastocyst stage when studied in different model
 - a. By treating mice with rotenone for 3 weeks before conception
 - b. By performing IVF at 20% vs 5%
 - c. In obese mice vs lean mice
 - d. In aged mice vs young mice
3. The elongation deficit is correctable by use of BGP-15, MitoQ and metformin, molecules that counteract the oxidative stress
4. The telomere defects persist after organogenesis and shorter telomeres persists in certain tissues (liver and heart of mice exposed preconceptionally to rotenone)
5. Interestingly there was no differences in telomere length between in vivo flushed embryos and embryos generated by IVF (in optimal conditions). Further longer embryo culture (CD5 or CD6) or vitrification did not affect telomere length.

The authors conclude that maternal signals are transmitted to the next generation during the peri-conception period.

Specifically, during the hours between ovulation and syngamy, zygotic mitochondria respond acutely to maternal physiological or in vitro exogenous cues, signal to the embryonic pronuclei and 'program' the capacity for telomere elongation between the 8-cell and ICM formation. Further, the identification of specific pharmaceutical compounds that can modulate these processes, offer therapeutic opportunities

The authors must be complemented for the extensive experimental work conducted and the valuable information obtained. Further, this work has valuable translational application for obese and aging individuals and for the field of IVF, given that shorter telomeres have been reported in IVF offspring.

Several comments follow

1. While the pronuclear transfer experiments offer important clues as the PN rather than the cytoplasm as the initial locus of regulation of telomere length, the nature of the signal(s) is not studied. Further, the statement (line 223) "disrupted

mitochondrial function in the oocyte at fertilization causes pronuclear epigenetic changes prior to syngamy that are responsible for impaired telomere elongation in ICM at the blastocyst stage” or the conclusion (line 392-394) seem to suggest the opposite, that the mitochondrial and cytoplasm are the initial locus of the initial stress. This needs better clarification. Further, “mitochondria” and “cytoplasm” seem to be used interchangeably and this is confusing.

2. BGP-15 is a complex molecule and PARP inhibitor whose mechanism of action is complex and multifarious. Further experiments separating the effect of this molecule compared to MitoQ alone would address the role of oxidative stress in the process.
3. Figure S4: more experiments are needed to show if the shorter telomeres found in embryos generated by IVF in 20% oxygen persist in fetuses (or offspring)
4. The discussion should emphasize that IVF performed in optimal conditions, culturing embryos to CD6 or vitrification in mouse models do not affect telomere length. This is important given the wide use of ART technologies

Reviewer #2

(Remarks to the Author)

In this manuscript, Winstanley et al propose that “telomere length is determined by mitochondrial-nuclear communication at fertilization”.

To reach this conclusion, the authors assess several measurements during preimplantation embryo development in mice which include telomere length, mitochondrial activity and pronuclear reprogramming (based on 5hmC staining). These measurements were taken at different stages and under different conditions (O₂ concentration, age and obese) which promote oxidative stress. While the observations made in the study are of great potential interest, especially given the rescue experiments, this reviewer has a number of issues/questions with it that should be address before recommending its publication:

1. Are the defects observed (mitochondrial failure and telomere length) are truly linked or just concomitant (=independent but coinciding in time) during the stress response? To address this, the authors perform the beautiful pronuclei/cytoplasm transfers to separate mitochondria from (pro)nuclear effects (Fig. 3). Precisely, the telomere length does not change when the cytoplasm has been “rotenone-treated”. I believe this certainly indicates that telomere length is independent from cytoplasm components (mitochondria) or communication between cytoplasm and nucleus. Am I missing something here?
2. Regarding the telomere length: in general, it decreases in oocytes from the conditions assayed, stay unchanged in 8C embryos and then decrease again in blastocysts (ICM). I find this quite surprising. If only a single mechanism is involved in the process, this would entail that the mechanism is disrupted in the oocytes and ICM, but not in the middle stage (?). Alternatively, there could be 2 mechanisms involved in early and late preimplantation development: one maternal and another zygotic given that zygotic genome activation (ZGA) occurs around 2C-stage. I wonder, is there a simple way to test if ZGA is required for the decrease in telomere length in blastocysts? Maybe blocking ZGA in embryos cultured in 5% or 20% and measure telomere length? I’d like to make clear that I’m not requesting the author to make this experiment (if they do it, great). Rather I’d like to know if the authors have considered this possibility (2 mechanisms, maternal and zygotic) to explain their surprising results in telomere length depending on developmental stage.
3. Mitochondrial content has been assessed using MitoTracker (5% vs 20% O₂ and control vs rotenone) only in zygotes (?), stage is not clear in figure legend). Given the known changes in mtDNA synthesis and degradation (McConnell & Petrie, 2004), and maybe mitochondrial mass (?), during preimplantation embryo development and TE vs ICM (Kelly et al., 2012), it’d be good to assess the mass (or copy number better) at different stages/different conditions. This would improve the study. Alternatively, the authors could comment on this, maybe in the discussion. Kelly, R. D., Mahmud, A., McKenzie, M., Trounce, I. A., & St John, J. C. (2012). Mitochondrial DNA copy number is regulated in a tissue specific manner by DNA methylation of the nuclear-encoded DNA polymerase gamma A. *Nucleic Acids Research*, 40, 10124–10138.

McConnell, J. M., & Petrie, L. (2004). Mitochondrial DNA turnover occurs during preimplantation development and can be modulated by environmental factors. *Reproductive Biomedicine Online*, 9, 418–424.

4. There are also a number of technical issues and inconsistencies in the graphs that should really be corrected to improve the manuscript and make the most out of the obtained results:

- a. Regarding the telomere length measurement: how is it really measured? What are the units? These are missing in the figures. Fig 1A indicates it is the ratio telomere/Rn18S, which ratio? $\Delta\Delta C_t$ ’s ratios, C_t ’s ratios,...?
- b. Why was that specific reference gene selected as opposed to others used in the literature like 36B4. Information on this regard should be included.
- d. Why are NAD(P)H and FAD⁺⁺ measurements done on a single z-plane from the z-stack rather than from the whole z-projection like in the MitoTracker and MMP assays? Using single z-plane add extra variability that should be avoided, specially when the treatments applied to females might affect oocyte/embryo shape.
- e. Specific information on the embryo stage should be included in all figures. This is not always the case, for example in Fig 1J-M, this is not clear.
- f. Specific information about number of embryos used in each assay should be included.
- g. It is really surprising to have d5 and d6 blastocysts when mouse embryos implant at day 4.5pc. Please explain.
- h. Rather than using violin plots AND scatterplot, the authors should present the data with scatterplots: they better represent the variability between embryos. Is there a reason for choosing the two types?
- i. It is confusing to use sometimes letters, sometime “*” to indicate statistically significant results. Please homogenise it. More so when the meaning of the letters is not really explained in the figure legends.
- j. Regarding the statistical tests: did the authors check for the normality of the data? T-test assumes that data are normally distributed, and they do not seem to be. The same for the ANOVA’s. If data are not normally distributed, non-parametric tests should be performed.

Minor comments:

1. Figure S3 and S4A are not mentioned in the text.
2. Was the telomere length measured in foetal kidneys from rotenone treated embryos gestated in surrogate females?
3. Indicate colour legend in all confocal/spinning disc images.

Reviewer #3

(Remarks to the Author)

The paper by Robker and colleagues described the effect of differences in oxygen concentrations on telomere length changes of the early mouse embryo. Initial work by the Rudolph group had established that telomeres elongate before the ICM stage of the preimplantation blastocysts. The authors expose female mice of different age or weight to agents that alter oxidative stress prior to fertilization. The authors find that telomere length increases are reduced under oxidative stress conditions. This effect can be restored by exposure to agents that reduce oxidative stress. The presented paper presents a large amount of work, yet remains largely phenomenological without mechanistic insights. Moreover, I see several complications with the interpretation of the presented data.

The authors use a q-PCR assay to quantify telomere length. This assay is highly questionable to determine telomere length changes and the authors need to demonstrate their findings with a better and more accurate assay. It is important to show the absolute telomere length and the telomere length distributions. This can be done using now established nanopore sequencing protocols.

As the authors point out, their nuclear transfer experiments demonstrate that the telomere length defect is caused by a nuclear encoded defect. The defect can not be reset by non-exposed cytoplasm. This finding needs to be addressed as it dissociates the timing of when the oxidative damage/stress occurs and when telomeres are elongating. A simple interpretation is that telomeres have been damaged by the treatment to the point that telomere elongation by telomerase is not permissive. This would be in line with previous work (for example see PMID: 29604323).

The authors should also follow up on their findings mentioned in line 130 to 133 of the manuscript. Robust embryo transfer protocols in normO₂ exist, 20+ year knowledge that TERT deficient mice only show fecundity defects in late stage intercrosses, and TERT KO mice have been successfully used in nuclear transfer and embryo transfer experiments. It would be important to reconcile these observations with the defects mentioned by the authors.

In addition I would like to urge the authors, particularly those with expertise in telomere biology, to present a more balanced introduction about the effects of telomere length on human health. The adverse effects of long telomeres are by now well established (e.g. PMID: 37140166). Short telomere phenotypes are largely restricted to a small population with genetic defects causing their telomere length to range below that 1%tile of the general population. Most of the references provided by the authors that link short telomeres in the general population to adverse outcomes are phenomenological or underpowered. An equal number of papers could be cited with opposing claims.

Version 1:

Reviewer comments:

Reviewer #1

(Remarks to the Author)

the authors have responded satisfactorily to the reviewer requests

Reviewer #2

(Remarks to the Author)

The revised version of "Telomere length in offspring is determined by mitochondrial-nuclear communication at fertilization" has greatly improved. Not only more experimental prove has been provided but also the format, style and writing has been polished, making the reading easier and conclusions better explained.

I have to admit that I'm impressed by the amount of different treatments used to prove the authors hypothesis, but also by the embryo/nuclear transfer experiments to further assess the origin of the defects in telomere lengthening.

I'm completely satisfied by the answers provided by the authors to my previous comments. Hence, I do recommend this paper to be accepted in its present version.

Minor comments:

There are small typos and details that should be corrected in the final version:

-In the figure panels, small letters are used to identify them, but in the figure legends capital letters are used. In Fig. 5, in the panels, F is used, but in the rest, small letters are used.

-Fig. S2: please check the stats significance. "(a; * p<0.05, **** p<0.0001) or linear mixedeffects model (c-f, where different letters indicate p<0.05)". b is missing (?) and f is nowhere to be seen.

-The numbering of the supplementary figures is somewhat odd: they do not appear in order in the main manuscript. See Fig. S3 as an example.

Reviewer #3

(Remarks to the Author)

The additional data provided by the authors did increase my concern that the authors are looking at an artifact that is caused by using qPCR as a tool to analyze telomere length.

Here are the main issues:

-The new data provided in figure 1J does not reflect the established telomere length seen in mice.

- an average telomere length loss of >70kb dramatically exceeds telomere length changes that are compatible with the telomere reserve present at the majority of mouse chromosome ends and inconsistent with the phenotype of the TERT KO mouse.

- The authors use inbred mouse strains. It is well established that the telomere length differences within littermates of such inbred strains is much smaller than the telomere length differences reported by the authors. The large variations seen in the measurements within each experimental group thus provides a good estimate of the experimental accuracy of the qPCR assay.

Without an additional assay to measure telomere length the work is inappropriate for publication.

Version 2:

Reviewer comments:

Reviewer #3

(Remarks to the Author)

The revision does not address my concern. The authors use a method to measure telomere length that has been shown by multiple studies to be highly inaccurate and to generate significant artifacts. While I appreciate that the authors attempt to validate their methodology, almost all key experiments rely on this PCR based method to measure telomere length.

Reviewer #4

(Remarks to the Author)

In this manuscript, the authors describe a significant effect of mitochondrial activity on telomere elongation between the 8-cell and blastocyst stages. Based on this data, the authors suggest that it is possible to modulate the preimplantation telomere resetting process using mitochondrially-targeted therapeutics (BGP-15, MitoQ, SS-31, metformin).

The principal concern with this study pertains to the use of qPCR to measure changes in telomere length. The authors do propose an alternative method for measuring telomere length, but all the key findings rely solely on qPCR. Coupled with the low amount of DNA used in most of these studies, the risk of artefacts is high. Whilst I appreciate that alternative methods for measuring telomere length are challenging in this context, I remain sceptical about the data presented in this study. The levels of telomere elongation observed are not compatible with the phenotypes described in telomerase-deficient mice.

A crucial experiment that would lend confidence to this study would be the analysis of telomerase-deficient mice. Examining changes in telomere length in the presence or absence of mitochondrial modulators would bolster confidence in the data provided. Without additional controls, I remain exceedingly sceptical of the changes in telomere length reported in this manuscript.

Version 3:

Reviewer comments:

Reviewer #3

(Remarks to the Author)

My fundamental concern with this work remains unresolved. The authors fail to convincingly establish that the measured telomere length changes in this paper are physiologically meaningful. This issue arises primarily because the work relies on a telomere length measurement tool that has been demonstrated to produce artifacts. Despite several rounds of revision, this criticism has not been adequately addressed. The piecemeal approach of iteratively making incremental changes to the telomere length measurements has not yielded data that support the broad claims made in this paper.

Reviewer #4

(Remarks to the Author)

The revised manuscript fails to address my previous. In my previous revision, I suggested additional experiments to address the issues associated with qPCR as a method to measure telomere length. Specifically, I suggested:

To use an alternative method for telomere length measurement:

While the authors have included FISH quantification to support the qPCR data, this addition remains insufficient. FISH quantification without appropriate controls—such as cell lines with known telomere lengths as standards—makes the interpretation of the data extremely challenging. Without these controls, the reliability of the FISH measurements cannot be adequately assessed.

Inclusion of telomerase KO mice as controls: I suggested to include telomerase KO mice as a control aimed validating the reported changes in telomere length. The authors decided not to perform these experiments since “previous reports have already described the essential role of telomerase in embryo telomere elongation”. This is reasoning is perplexing: this is precisely the reason, I suggested including this control.

Point-by-point response to the reviewers' comments

We thank the editorial team for the invitation to submit a revision; and now provide a substantially revised manuscript that includes a large amount of additional experimental work. A total of 22 new data panels (15 in the main Figures and 7 in the Supp Figures) are included in this revised manuscript expanding the number of Figures from five to nine. The new data panels are outlined with red boxes in the Tracked Changes document. This new data greatly strengthens the study by providing additional evidence that supports our original conclusions. As requested, mechanistic details are provided that more strongly show:

- 1) mitochondria are the initial source of the signal;
- 2) 8-oxodG modifications and/or changes to DNA methylation are candidate mechanisms by which the mitochondrial signal is transmitted to the nucleus;
- 3) bi-phasic nature of the mechanism which occurs within the first 8h following fertilization to influence telomere elongation capacity in the subsequent days of embryogenesis.

Reviewer #1 Comments and Responses

The work titled "Telomere length in offspring is determined by mitochondrial-nuclear communication at fertilization" studies the effects of oxidative stress 1) preconception (with rotenone given for 3 weeks prior to conception), 2) with IVF performed under optimal (5%) or high (20%) oxygen, 3) in aged (one year vs 3-4 months) or 4) obese female mice on embryonic (including ICM) telomere length, mitochondrial mass and membrane potential, global 5m and 5hm and offspring telomere length. The authors also added several molecules to counteract these effects, specifically BGP-15 (an antioxidant and PARP inhibitor), MitoQ and metformin.

In addition, pronuclear transfer experiments were performed to evaluate how the PN or mitochondria affect telomere length.

The authors found that:

1. mitochondrial-nuclear communication at fertilization controls telomere elongation during formation of the pluripotent Inner Cell Mass (ICM). In particular (fig 2B), PN transfer of rotenone exposed PN into control cytoplasm resulted in shorter ICM telomeres, while transfer of control PN into rotenone exposed cytoplasm did not shorten the telomeres, suggesting that the PN is the locus responsible to regulate telomere elongation
2. Oxidative stress reduces the telomere elongation that occurs at the morula and blastocyst stage when studied in different model
 - a. By treating mice with rotenone for 3 weeks before conception
 - b. By performing IVF at 20% vs 5%
 - c. In obese mice vs lean mice
 - d. In aged mice vs young mice
3. The elongation deficit is correctable by use of BGP-15, MitoQ and metformin, molecules that counteract the oxidative stress
4. The telomere defects persist after organogenesis and shorter telomeres persists in certain tissues (liver and heart of mice exposed preconceptionally to rotenone)
5. Interestingly there was no differences in telomere length between in vivo flushed embryos and embryos generated by IVF (in optimal conditions). Further longer embryo culture (CD5 or CD6) or vitrification did not affect telomere length.

The authors conclude that maternal signals are transmitted to the next generation during the peri-conception period. Specifically, during the hours between ovulation and syngamy, zygotic mitochondria respond acutely to maternal physiological or in vitro exogenous cues, signal to the embryonic pronuclei and 'program' the capacity for telomere elongation

between the 8-cell and ICM formation. Further, the identification of specific pharmaceutical compounds that can modulate these processes, offer therapeutic opportunities

The authors must be complemented for the extensive experimental work conducted and the valuable information obtained. Further, this work has valuable translational application for obese and aging individuals and for the field of IVF, given that shorter telomeres have been reported in IVF offspring.

Several comments follow

Response:

We thank Reviewer 1 for their complimentary review of our manuscript: 'extensive experimental work', 'valuable information', and 'valuable translation application'. We have responded to each of their follow-up comments below and incorporated their suggested changes into the manuscript.

R1 Comment 1: While the pronuclear transfer experiments offer important clues as the PN rather than the cytoplasm as the initial locus of regulation of telomere length, the nature of the signal(s) is not studied. Further, the statement (line 223) "disrupted mitochondrial function in the oocyte at fertilization causes pronuclear epigenetic changes prior to syngamy that are responsible for impaired telomere elongation in ICM at the blastocyst stage" or the conclusion (line 392-394) seem to suggest the opposite, that the mitochondrial and cytoplasm are the initial locus of the initial stress. This needs better clarification. Further, "mitochondria" and "cytoplasm" seem to be used interchangeably and this is confusing.

Response:

We agree with the Reviewer that the 'initial locus of the initial stress' is the mitochondria/cytoplasm, and that the pronuclear changes are downstream of mitochondrial disruption. We have re-written the paragraph that describes the pronuclear transfer experiments (lines 229-237 Tracked document), as suggested by the Reviewer, to more carefully explain these findings. More importantly, we have conducted a number of additional experiments to strengthen this conclusion.

To further test whether mitochondrial activity is the initial locus of the initial stress, we conducted a series of new experiments that treated rotenone-stressed eggs with three different mitochondria-targeted drugs from fertilization to 2PN formation (8 hours). The results confirm that modulating mitochondrial activity during this narrow window of the first cell cycle leads to improved telomere elongation 4 days later at the blastocyst stage. **New data: Fig. 6a-6g.**

To further examine 'the nature of the signal(s)' we provide new data that 8-oxodG is a likely mechanism by which the mitochondria, via mtROS, communicate to the nucleus. Thus we now demonstrate two potential nuclear modifications (8-oxodG and hydroxymethylation) by which embryo pronuclei could influence subsequent telomere elongation. **New data: Fig 2g, Fig 4f, Fig S11a, Fig S12b.**

R1 Comment 2: BGP-15 is a complex molecule and PARP inhibitor whose mechanism of action is complex and multifarious. Further experiments separating the effect of this molecule compared to MitoQ alone would address the role of oxidative stress in the process.

Response:

We have conducted additional experiments directly comparing the effects of BGP-15 and MitoQ. To further address the role of oxidative stress we also included SS-31 a small tetra-peptide that directly binds cardiolipin (phospholipid specific to mitochondria) and reduces mtROS. The results show that each of these three compounds reduces mtROS in zygotes within 8h post-fertilization, with subsequent improvement of telomere elongation in ICM (see new data Fig 6a-6g). That each of these three compounds causes the exact same effects strengthens the evidence that oxidative stress, specifically mtROS, are part of the signal from mitochondria to the nucleus that regulates embryo telomere length. **New data: Fig. 6a-6g.**

R1 Comment 3: Figure S4: more experiments are needed to show if the shorter telomeres found in embryos generated by IVF in 20% oxygen persist in fetuses (or offspring)

Response:

We have analyzed the fetal tissues from this experiment. However, there are considerable differences between fetuses from the two groups that confounds any direct comparisons. Firstly, embryos cultured at 20% O₂ exhibited reduced implantation potential compared to those cultured at 5% O₂ (Fig S4), meaning that the most severely affected embryos are not available for analysis. In addition, fetuses from the 20% O₂ group were significantly heavier (data not shown) likely due to the reduced litter size, yet indicative of an altered growth trajectory in this group. Analysis of telomere length in the tissues of the surviving fetuses found no differences between the groups but it is entirely possible that the telomeres of the fetuses that were most affected (and thus not viable) had shorter telomeres, similar to the blastocysts. We have now included information in the manuscript (see below) to indicate that telomere lengths were not different in the surviving fetuses, with the caveat that the fetuses were not equivalent and therefore fetal tissues should not be directly compared.

New text (lines 87-88 Tracked document): 'Thus telomere lengths in offspring could not be compared in equivalent fetal tissues, due to the confounding effect of fetal loss in one group.'

New text (Fig S4 legend): 'Analysis of telomere length in tissues (liver, kidney, heart) of the surviving pups found no significant difference between groups (data not shown).'

R1 Comment 4: The discussion should emphasize that IVF performed in optimal conditions, culturing embryos to CD6 or vitrification in mouse models do not affect telomere length. This is important given the wide use of ART technologies

Response:

We thank the Reviewer for highlighting this issue and have added this point to the first paragraph of the Discussion.

New text (lines 358-362 Tracked document): 'Reassuringly, common assisted reproduction techniques did not influence telomere elongation in our model. Specifically, ICM telomere length was not altered by IVF and culture in optimized conditions compared to in vivo-conceived embryos (Supplementary Fig. S3b), by extended embryo culture to day 6 (Fig. 1e), or by vitrification at the morula stage (Supplementary Fig. S4a, S9a).'

Reviewer #2 Comments and Responses

In this manuscript, Winstanley et al propose that “telomere length is determined by mitochondrial-nuclear communication at fertilization”.

To reach this conclusion, the authors assess several measurements during preimplantation embryo development in mice which include telomere length, mitochondrial activity and pronuclear reprogramming (based on 5hmC staining). These measurements were taken at different stages and under different conditions (O₂% concentration, age and obese) which promote oxidative stress. While the observations made in the study are of great potential interest, especially given the rescue experiments, this reviewer has a number of issues/questions with it that should be address before recommending its publication:

Response:

We thank Reviewer 2 for their positive remarks about the manuscript: ‘the observations are of great potential interest’, and ‘the beautiful pronuclei/cytoplasm transfers to separate mitochondria from (pro)nuclear effects’; and for their excellent suggestions on how to clarify the conclusions. We have responded to each of their comments below and made the suggested changes in the manuscript.

R2 Comment 1: Are the defects observed (mitochondrial failure and telomere length) are truly linked or just concomitant (=independent but coinciding in time) during the stress response? To address this, the authors perform the beautiful pronuclei/cytoplasm transfers to separate mitochondria from (pro)nuclear effects (Fig. 3). Precisely, the telomere length does not change when the cytoplasm has been “rotenone-treated”. I believe this certainly indicates that telomere length is independent from cytoplasm components (mitochondria) or communication between cytoplasm and nucleus. Am I missing something here?

Response:

We have 1) revised our description of this experiment for clarity and 2) conducted additional experiments that use more specific mitochondrial-targeted compounds to show that mitochondrial failure and telomere length are linked. The results confirm that the mitochondrial influence on telomere elongation mechanisms is restricted to the first 8h following fertilization.

- 1) Rotenone is a highly specific disruptor of mitochondrial Complex I. That the pronuclei alone from the rotenone-treated zygotes transmit the defect for shortened telomeres indicates that a mitochondrial ‘damage’ signal has been transmitted to the DNA. In contrast, when ‘non-exposed’ nuclei are placed into rotenone cytoplasm, telomere elongation proceeds normally. This demonstrates that the mito-nuclear signal occurs specifically during the hours prior to syngamy, and that the presence of damaged mitochondria after this early developmental window does not have the same effect. We have clarified our explanation of these findings.

New text (Lines 229-237): ‘This indicates that the mitochondrial damage present in rotenone-exposed zygotes altered the nuclear DNA in a manner that resulted in impaired telomere elongation in ICM. Thus, the nuclear material of rotenone-exposed zygotes contains the molecular signals leading to later impaired telomere elongation potential. Conversely, reconstructed embryos that received cytoplasm of rotenone-exposed zygotes did not have reduced telomere length in ICM (Fig. 6j). This

indicates that the signaling from damaged mitochondria to nuclear DNA, that impairs telomere elongation, occurs prior to syngamy; and that the presence of damaged mitochondria following the pronuclear stage does not impact this process.'

- 2) The new additional experiments use specific mitochondrial-targeted compounds in a highly restricted window to further show that 'mitochondrial failure' and telomere length are linked. Specifically, rotenone-exposed oocytes are treated with either BGP-15, MitoQ or SS-31 (a cardiolipin-binding peptide) for just 8h and each of these three compounds reduced zygote mtROS and improved ICM telomere length. This new data demonstrates that targeted mitochondrial disruption restricted to the time between fertilization and syngamy is responsible for impaired telomere elongation.
New data: Fig. 6a-g.

R2 Comment 2: Regarding the telomere length: in general, it decreases in oocytes from the conditions assayed, stay unchanged in 8C embryos and then decrease again in blastocysts (ICM). I find this quite surprising. If only a single mechanism is involved in the process, this would entail that the mechanism is disrupted in the oocytes and ICM, but not in the middle stage (?). Alternatively, there could be 2 mechanisms involved in early and late preimplantation development: one maternal and another zygotic given that zygotic genome activation (ZGA) occurs around 2C-stage. I wonder, is there a simple way to test if ZGA is required for the decrease in telomere length in blastocysts? Maybe blocking ZGA in embryos cultured in 5% or 20% and measure telomere length? I'd like to make clear that I'm not requesting the author to make this experiment (if they do it, great). Rather I'd like to know if the authors have considered this possibility (2 mechanisms, maternal and zygotic) to explain their surprising results in telomere length depending on developmental stage.

Response:

We absolutely agree with the Reviewer's assessment that two mechanisms are involved. Our observations of modest telomere elongation prior to 8-cells and greater elongation between 8-cells and blastocyst are consistent with the literature showing that putative ALT-like mechanisms occur soon after fertilization and that telomerase-mediated mechanisms occur between the morula to blastocyst stage.

As further evidence, we have included new data (see Supp Fig S2c-f) showing that in each of four additional strains of mice the most consistent phase of telomere elongation is between 8-cells and blastocyst. This new data shows even more clearly that telomere elongation occurs in two phases, and the predominant, and most consistent, phase is the one is between 8C and ICM.

New data: Supp Fig S2c-f. New text (lines 61-63 Tracked document): 'Telomere elongation between IVF and blastocyst formation was similarly examined in 4 additional strains of mice and consistently showed the greatest increase in telomere length only between 8-cells and blastocysts (Supplementary Fig. S2c-f).'

Indeed, an exciting finding of our study is that mitochondrial dysfunction/ oxidative stress distinctly impacts only the later stage of telomere elongation ie the phase between 8-cells and blastocyst. We have clarified this concept of the two phases with a new paragraph in the Discussion.

New text (lines 387-400 Tracked document): 'Because our experiments measured telomere length at multiple points during preimplantation development, they revealed that distinctly different regulatory mechanisms control the first phase of telomere elongation compared to the second phase. Specifically, in each of the models of mitochondrial disruption (20% O2 culture, rotenone, aging, obesity) telomere lengths were identical at the 8-cell stage, even in contexts where telomeres started out shorter

in ovulated oocytes, i.e. with rotenone-exposure, aging or obesity. This indicates that the earliest phase of telomere elongation, which is thought to be mediated via the recombination-based Alternative Lengthening of Telomeres (ALT) mechanism (11,12), is not affected by the defects that are present in these oocytes. In contrast, each of the models of mitochondrial dysfunction exhibited the exact same defect in telomere elongation between the 8-cell stage and the ICM of the blastocyst. This phase of telomere elongation is known to be regulated by telomerase-mediated mechanisms (11-13), and suggests that mitochondrial dysfunction, and specifically increased mtROS, interferes with the normal functioning of this pathway.'

We agree with the Reviewer that there maternal and zygotic mechanisms at play. The suggestion to test whether ZGA is required is an interesting one. However, blocking ZGA would almost certainly prevent the induction of telomerase and thus telomere elongation would be prevented due to this defect.

R2 Comment 3: Mitochondrial content has been assessed using MitoTracker (5% vs 20% O₂ and control vs rotenone) only in zygotes (? , stage is not clear in figure legend). Given the known changes in mtDNA synthesis and degradation (McConnell & Petrie, 2004), and maybe mitochondrial mass (?), during preimplantation embryo development and TE vs ICM (Kelly et al., 2012), it'd be good to assess the mass (or copy number better) at different stages/different conditions. This would improve the study. Alternatively, the authors could comment on this, maybe in the discussion.

Response:

The Reviewer is correct- Mitotracker (mitochondrial content) was analyzed in zygotes. We have now labelled the Figure (Fig 2e) as such and updated the figure legend. However, there was not a strong rationale to investigate mitochondrial mass/ Mitotracker at later developmental timepoints once our data (ie Fig 6) demonstrated that the mitochondrial influence on telomere elongation occurred only at the zygote stage and not later.

The Reviewer also suggested measuring mtDNA copy number, as an even better marker. We have in fact measured mtDNA copy number across preimplantation development in each of the oxidative stress contexts that we studied (20% O₂, rotenone, obesity, aging). Our original hypothesis going into this research area was that mtDNA replication and telomere elongation might be linked. However, we found this is not the case. We find no correlations between telomere length and mtDNA copy number at any stage of embryogenesis. In contrast, what we did find (as described in this manuscript) was that mtROS at the zygote stage was tightly associated with capacity for telomere elongation at the blastocyst stage.

Our extensive data on mtDNA copy number during embryogenesis was not included in this manuscript because we could find no link to telomere elongation. This data, although not informative to the current study, can be provided to the Reviewer if requested by the Editor.

R2 Comment 4: There are also a number of technical issues and inconsistencies in the graphs that should really be corrected to improve the manuscript and make the most out of the obtained results:

R2 Comment 4a: Regarding the telomere length measurement: how is it really measured? What are the units? These are missing in the figures. Fig 1A indicates it is the ratio telomere/Rn18S, which ratio? $\Delta\Delta C_t$'s ratios, C_t 's ratios,...?

Response:

The Methods state that 'Telomere relative to Rn18S copies was calculated using the $2^{-\Delta\Delta C_t}$ method....' followed by further details of the calculation. These units are now specified in the legends of every Figure. We thank the Reviewer for this suggestion and the one below which help delineate the telomere assay of oocytes/embryos from that of the fetal tissue samples.

R2 Comment 4b: Why was that specific reference gene selected as opposed to others used in the literature like 36B4. Information on this regard should be included.

Response:

Rn18S was selected as the reference gene due to its multi-copy status. 36B4 is a common reference gene for telomere assays but it is a single-copy gene and thus cannot be accurately detected in samples with low nuclear DNA content, such as individual MII oocytes and 2-cell embryos. We have added an explanation into the methods section for "Telomere and Rn18S qPCR using DNA from individual oocytes and embryos". The specific reference gene used in each type of assay is now clarified in the Methods and legends.

New text (lines 602-604 Tracked document): 'Rn18S was used as the reference gene for all oocyte/ embryo samples because it is a multi-copy gene; and thus detectable in samples with low nuclear DNA, i.e. individual oocytes or 2-cell embryos'.

Note: there was no Reviewer comment 4c.

R2 comment 4d: Why are NAD(P)H and FAD⁺⁺ measurements done on a single z-plane from the z-stack rather than from the whole z-projection like in the MitoTracker and MMP assays? Using single z-plane add extra variability that should be avoided, specially when the treatments applied to females might affect oocyte/embryo shape.

Response:

Only morphologically normal embryos were analyzed and always imaged at the z-plane where the zygote diameter was the largest. We monitor oocyte size and have not found differences between treatment groups but the fluorescence measurements are normalized to oocyte area to control for this.

Autofluorescence imaging was performed on a different confocal (Olympus FV3000) to the MitoTracker and MMP assays (CV100 spinning disk), due to the strict requirements for emission detection bandwidths for NAD(P)H and FAD⁺⁺. Further, the FV3000 confocal images at a much slower speed than a spinning disk, therefore only one z-plane was imaged. Imaging a single z-plane is common in autofluorescence analyses and we followed these general protocols. For example: Adhikari D, et al., Depletion of oocyte dynamin-related protein 1 shows maternal-effect abnormalities in embryonic development. *Sci Adv.* 2022 Jun 17;8(24):eabl8070, or Tan TCY, et al. Optical imaging detects metabolic signatures associated with oocyte quality. *Biol Reprod.* 2022 Oct 11;107(4):1014-1025.

R2 comment 4e: Specific information on the embryo stage should be included in all figures. This is not always the case, for example in Fig 1J-M, this is not clear.

Response:

The embryo stage analyzed is now clearly stated in all Figures and associated legends.

R2 comment 4f: Specific information about number of embryos used in each assay should be included.

Response:

The number of embryos used in each assay is now clearly stated in each Figure legend.

R2 comment 4g: It is really surprising to have d5 and d6 blastocysts when mouse embryos implant at day 4.5pc. Please explain.

Response:

Thank you for pointing out this potentially confusing nomenclature used in Fig 1e. **We have clarified this important issue in the Methods and in the text of the Results describing Fig 1e.** IVF embryos are typically referred to by their “day of culture” and/or time in hours post-fertilization rather than days post-fertilization. For example, a zygote is ‘day 1 of culture’ (and not ‘day 0’), a 2-cell embryo is observed at 24 hours post-fertilization which is also ‘day 2 of culture’, and blastocysts are observed at 96 hours post-fertilization which is also ‘day 5 of culture’. The Reviewer is correct that mouse embryos conceived in vivo implant at 4.5pc; and an in vivo fertilized zygote is referred to as 0.5dpc. Thus, the “day 5” IVF blastocyst is the equivalent of a d4.5 in vivo embryo, at the time of implantation.

An IVF blastocyst at 96h post-fertilization = Day 5 of culture = 4.5 dpc in vivo (time of implantation)

An IVF blastocyst at 120h post-fertilization = Day 6 of culture = 5.5 dpc in vivo (post-implantation)

Clarified text (Lines 58-61 Tracked document): There was no further change in telomere length in immuno-purified ICM (see example Fig. 1d) between day 5 of culture (96h post-fertilization) which corresponds to E4.5 (the timeframe of normal implantation) and day 6 of culture i.e., E5.5 in vivo (Fig. 1e).

R2 comment 4h: Rather than using violin plots AND scatterplot, the authors should present the data with scatterplots: they better represent the variability between embryos. Is there a reason for choosing the two types?

Response:

Violin plots are only used for the telomere length graphs. They provide better visual clarity on the population distribution and are consistent with the standard of many reports in the literature. Scatterplots were used for all other graphs. This has been clarified in the statistics section of the Methods.

Clarified text (line 832-833 Tracked document): Telomere length data is displayed as mean \pm SEM overlaid on a violin plot to illustrate the distribution of data points, with the n-values indicated in the legend.

R2 comment 4i: It is confusing to use sometimes letters, sometime “*” to indicate statistically significant results. Please homogenise it. More so when the meaning of the letters is not really explained in the figure legends.

Response:

Letters were only used for data where multiple groups are compared, i.e a time-course as in Figures 1b, 2f, 4i. The legends now provide the exact p-values represented by the different letters. In addition, where statistical significance is indicated with an asterisk,

the P-value is provided in the legend. This sentence has also been added to the Statistics section of the Methods (lines 840-841).

R2 comment 4j: Regarding the statistical tests: did the authors check for the normality of the data? T-test assumes that data are normally distributed, and they do not seem to be. The same for the ANOVA's. If data are not normally distributed, non-parametric tests should be performed.

Response:

Yes- our data was analyzed by a statistician prior to the original manuscript submission. The normality check determined that the telomere data required log-transformation for analysis, as also noticed by the Reviewer. This is stated in the Materials and Methods section "Statistical Analysis" (lines 834-835): 'Analysis of normality showed that telomere data required log transformation for statistical analysis'.

R2 Minor comments:

R2 Minor comment 1: Figure S3 and S4A are not mentioned in the text.

Response:

These Figures are referred to in the Materials and Methods section. Figure S3a is described in "Telomere and Rn18S qPCR using DNA from individual oocytes and embryos". Figures S3b and S3c are described in "Derivation of oocytes, early embryos, and blastocysts". Figure S4a is described in "Transfer of blastocyst embryos to pseudo-pregnant female mice". Fig S3b and Fig S4a are also now mentioned in the first paragraph of the Discussion (lines 361-362 Tracked document).

R2 Minor comment 2: Was the telomere length measured in foetal kidneys from rotenone treated embryos gestated in surrogate females?

Response:

Yes- telomere length in fetal kidneys from the rotenone-exposed oocytes/ embryos gestated in surrogates was measured. This data is shown as Figure S9j.

R2 Minor comment 3: Indicate colour legend in all confocal/spinning disc images.

Response:

Colors have been defined in the legends.

Reviewer #3 Comments and Responses

The paper by Robker and colleagues described the effect of differences in oxygen concentrations on telomere length changes of the early mouse embryo. Initial work by the Rudolph group had established that telomeres elongate before the ICM stage of the preimplantation blastocysts. The authors expose female mice of different age or weight to agents that alter oxidative stress prior to fertilization. The authors find that telomere length increases are reduced under oxidative stress conditions. This effect can be restored by exposure to agents that reduce oxidative stress. The presented paper presents a large amount of work, yet remains largely phenomenological without mechanistic insights.

Moreover, I see several complications with the interpretation of the presented data.

Response:

We thank the reviewer for acknowledging the comprehensive nature of our work. In response to their feedback we have conducted a number of additional experiments – including measurements of absolute telomere length via a method that is feasible in embryos and examination of 8-oxodG modifications in zygotes. Each of the comments is addressed in detail below.

R3 Comment 1: Initial work by the Rudolph group had established that telomeres elongate before the ICM stage of the preimplantation blastocysts.

Response:

We believe the Reviewer is referring to the paper by Schaetzlein, Rudolph et.al (Proc Natl Acad Sci U S A (2004) 101:8034) which shows that telomere elongation in mouse embryos and bovine embryos occurs between the 8-cell/ morula stage and blastocyst stage. We agree this is seminal work and it has been foundational for our investigations in this topic. It is Ref #13 in the manuscript. (It was cited 4 times in the Introduction/Results and now an additional 2 times in the revised Discussion.) We have reproduced that telomere elongation occurs in mouse embryos between 8-cells/ morula and blastocyst (Fig 1b, 1h) and have built upon these observations to discover that this elongation mechanism is regulated by oocyte mitochondria at fertilization.

R3 Comment 2: The presented paper presents a large amount of work, yet remains largely phenomenological without mechanistic insights.

Moreover, I see several complications with the interpretation of the presented data.

Response:

Towards providing more mechanistic insights and solidifying the conclusions we have conducted additional experiments and have now added an additional 22 panels of data (15 panels in the main Figures and 7 in the Supplementary Figures). This new data confirms that there is multi-factorial regulation of telomere elongation in preimplantation embryos and we are committed to defining these mechanisms in future experiments; that can only now be designed based on the initial, highly novel findings of our current work. To reiterate the new mechanisms that we have identified in embryos: 1) telomere elongation during embryogenesis is regulated by mitochondrial ROS, 2) a mito-nuclear signal is transmitted in a highly restricted developmental timeframe (within 8h of fertilization), 3) the impairment to ICM telomere elongation has consequences for fetal tissue telomere length, 4) embryo telomere elongation defects are reversible. In response to the Reviewer's feedback, we have carefully considered our interpretations of the data and clarified these as requested throughout the manuscript.

R3 Comment 3: The authors use a q-PCR assay to quantify telomere length. This assay is highly questionable to determine telomere length changes and the authors need to demonstrate their findings with a better and more accurate assay. It is important to show the absolute telomere length and the telomere length distributions. This can be done using now established nanopore sequencing protocols.

Response:

With all due respect, it is not correct to assert that the assay is 'highly questionable'. We extensively validated the fidelity of our assay, confirmed that no PCR artefacts are generated, and demonstrate consistent results when compared to qFISH (Supp Fig S2a, S2b). Our observations of telomere length kinetics during pre-implantation development are consistent with those in the literature and across multiple lines of

mice (**new Supp Figs S2c-S2f**). Co-author Pickett has expertise in nanopore sequencing and molecular combing but these techniques are not possible with the limited DNA from embryos which are only ~40 cells (ICM) each. However, to provide additional evidence for our conclusions about differentials in telomere length, the key data has now been verified by an absolute telomere length assay as suggested by the Reviewer (new Figs 1j, 3d, 3f, 4k). These new results validate the relative telomere length assay and also provide a measure of absolute telomere length differences in the ICM (following either 20% O₂ or rotenone) and fetuses of mice fed rotenone versus controls. **New data: Fig 1j, Fig 3d, Fig 3f, Fig 4k.**

R3 Comment 4: As the authors point out, their nuclear transfer experiments demonstrate that the telomere length defect is caused by a nuclear encoded defect. The defect can not be reset by non-exposed cytoplasm. This finding needs to be addressed as it dissociates the timing of when the oxidative damage/stress occurs and when telomeres are elongating. A simple interpretation is that telomeres have been damaged by the treatment to the point that telomere elongation by telomerase is not permissive. This would be in line with previous work (for example see PMID: 29604323).

Response:

Yes- the Reviewer is correct. A key finding of this work is that the timing of oxidative damage/ stress (the zygote stage) is distinct from the timing of telomere elongation (between morula and blastocyst). To build on the nuclear transfer experiments, we now provide additional new data to demonstrate this point. Specifically, rotenone-exposed eggs were treated with one of three different mitochondrial-targeted drugs (BGP-15, MitoQ or SS-31) for just 8h and then cultured in standard media until the blastocyst stage. The results further demonstrate that oxidative damage/stress during the fertilization window causes the later defect in telomere elongation. **New data: Fig 6a-6g.**

We agree with the Reviewer that it is possible that telomeres are damaged by mtROS and that this impairs telomerase activity, a known mechanism in cancer cells. To address this comment, we provide new data that 8-oxodG modifications occur in zygotes in response to each type of oxidative stress (20% O₂, rotenone, aging, obesity). However, whether these DNA modifications impact telomerase activity or transcriptional mechanisms will require extensive investigations that we are actively pursuing but that are beyond the scope of the current manuscript's key findings (listed in R3 comment 2 response). Importantly, the oxidative-stress associated changes to zygote global methylation are also likely significant. Thus, the evidence indicates there is not a single simple interpretation as the Reviewer suggests; but rather there are multi-factorial epigenetic mechanisms linking mitochondrial activity at fertilization to telomere elongation capacity 3 days later. **New data: Fig 2g, Fig 4f, Fig S11a, Fig S12b.**

R3 Comment 5: The authors should also follow up on their findings mentioned in line 130 to 133 of the manuscript. Robust embryo transfer protocols in normO₂ exist, 20+ year knowledge that TERT deficient mice only show fecundity defects in late stage intercrosses, and TERT KO mice have been successfully used in nuclear transfer and embryo transfer experiments. It would be important to reconcile these observations with the defects mentioned by the authors.

Response:

It is not clear from the comment how our results conflict with these other observations.

1) Our 'normO₂' embryo transfer protocols were robust and resulted in many healthy fetuses, albeit at lower numbers than the 5% O₂ gold-standard. To our knowledge there has not been a previous direct comparison of post-implantation development following pre-implantation culture in 5% O₂ versus 20% O₂, even though several studies have shown that multiple aspects of pre-implantation development are compromised by culture in 20% O₂ (reviewed in *Reproduction* (2021) 161:F41).

2) Yes- TERT deficient mice show only fecundity defects in late-stage inter-crosses when oocyte telomeres have progressively become critically short to the point of impacting embryo viability. Our data reveals a distinctly different biological mechanism; showing for the first time that, even in genetically wildtype mice (with normal telomere length), maternal physiology influences embryo telomere length with consequences for offspring telomere length.

3) The reviewer may be referring to the work in which stem cells and cloned pups were created from telomerase deficient mice (selected refs below). Indeed we have followed this work which is extremely exciting because it demonstrates that telomere elongation capacity is possible when nuclei (even *Terc*^{-/-} nuclei) are transferred into a new ooplasm (ie somatic cell nuclear transfer (SCNT)). This is consistent with observations (select refs below) that telomere regeneration is more efficient in embryonic stem cells generated via SCNT compared to iPSC. To highlight this important implication of our study, which was downplayed in the original submission, we have added a new sentence to the Discussion.

Line 381 (Tracked document): Thus, our identification of this new connection between mitochondria and telomeres during preimplantation embryogenesis may also extend to other contexts of cellular reprogramming and stem cell biology. **New text:** 'In particular, this mechanism controlled by oocyte mitochondria may explain the enhanced telomere rejuvenation that occurs in embryonic stem cells generated by SCNT compared to iPSCs (51(*Cell Stem Cell*. 2014)).'

- Telomere elongation and naive pluripotent stem cells achieved from telomerase haplo-insufficient cells by somatic cell nuclear transfer *Cell Rep* 2014 Dec 11;9(5):1603-1609 PMID: 25464850
- Enhanced telomere rejuvenation in pluripotent cells reprogrammed via nuclear transfer relative to induced pluripotent stem cells *Cell Stem Cell*. 2014 Jan 2;14(1):27-39. PMID: 24268696

R3 Comment 6: In addition I would like to urge the authors, particularly those with expertise in telomere biology, to present a more balanced introduction about the effects of telomere length on human health. The adverse effects of long telomeres are by now well established (e.g. PMID: 37140166). Short telomere phenotypes are largely restricted to a small population with genetic defects causing their telomere length to range below that 1%tile of the general population. Most of the references provided by the authors that link short telomeres in the general population to adverse outcomes are phenomenological or underpowered. An equal number of papers could be cited with opposing claims.

Response:

We have addressed this point in the Introduction and the Discussion.

1. By far the weight of evidence in the literature is that shortened LTL (leukocyte telomere length) is associated with tissue dysfunction and adverse outcomes. In the Introduction only one supporting reference was cited; a robust meta-analysis of 27 studies demonstrating that shorter LTL was associated with stroke, myocardial infarction and type 2 diabetes (PMID: 25406241).

To provide further evidence we have now included an additional reference. This recent publication (***JAMA Intern Med* (2022) 182(3):291**) included >470K

individuals and demonstrates that shorter LTL is associated with a higher risk of specific types of organ-specific disease and mortality, specifically cardiovascular, digestive, respiratory, musculoskeletal and hematopoietic diseases.

Because, as perhaps the Reviewer is suggesting, leukocyte telomere length has not been demonstrated so clearly as a marker for all-cause mortality, we have also removed the word 'lifespan' from the first sentence in the Introduction.

2. In the Discussion, we have clarified that our work has uncovered a potential mechanistic basis for the natural variations in telomere length that occur in humans; and that it is not relevant for understanding the phenotypes caused by extremely rare genetic telomere disorders that result in clinical syndromes. This new text cites the reference suggested by the Reviewer (PMID: 37140166 and another about rare genetic defects causing short telomere phenotypes).

New text (lines 365-368 Tracked document): 'Importantly, these changes in telomere length are distinct from those caused by rare genetic disorders that result in profound telomere length defects and clinical syndromes (a short ref and a long ref), and instead provide a potential mechanistic basis for the natural variations in telomere length that occur in humans.'

Point-by-point response to Reviewer Comments.

We have addressed each of the Reviewer Comments in full, as detailed below.

REVIEWER COMMENTS

Reviewer #1 (Remarks to the Author):

the authors have responded satisfactorily to the reviewer requests

Response: We thank the Reviewer for supporting the publication of our work.

Reviewer #2 (Remarks to the Author):

The revised version of “Telomere length in offspring is determined by mitochondrial-nuclear communication at fertilization” has greatly improved. Not only more experimental prove has been provided but also the format, style and writing has been polished, making the reading easier and conclusions better explained.

I have to admit that I'm impressed by the amount of different treatments used to prove the authors hypothesis, but also by the embryo/nuclear transfer experiments to further assess the origin of the defects in telomere lengthening.

I'm completely satisfied by the answers provided by the authors to my previous comments. Hence, I do recommend this paper to be accepted in its present version.

Response: We thank the Reviewer for supporting the publication of our work.

Minor comments:

There are small typos and details that should be corrected in the final version:

-In the figure panels, small letters are used to identify them, but in the figure legends capital letters are used. In Fig. 5, in the panels, F is used, but in the rest, small letters are used.

Response: the labelling of Fig 5 panel f has been corrected. The legends have been corrected to use lowercase letters.

*-Fig. S2: please check the stats significance. “(a; * $p < 0.05$, **** $p < 0.0001$) or linear mixed effects model (c-f, where different letters indicate $p < 0.05$)”. b is missing (?) and f is nowhere to be seen.*

Response: The 'letters' are referring to the Figure panel and not the statical differences between groups. This has been clarified.

Edited text (Supp Fig 2 legend): Data analyzed using one-way ANOVA (panels a,b; * $p < 0.05$, **** $p < 0.0001$) or linear mixed-effects model (panels c-f, where different letters indicate $p < 0.05$).

-The numbering of the supplementary figures is somewhat odd: they do not appear in order in the main manuscript. See Fig. S3 as an example.

Response: We have included a significant amount of material in the Supplemental Figures to show, as completely as possible, how the experiments were conducted and important control experiments that were performed. Supp Fig 3 shows that IVF embryos have normal telomere length and that male and female embryos are not different. These are important issues for embryologists but did not merit highlighting in the main text. Thus Supp Fig 3 is not described until the Methods section.

Edited Figure and text: We have rearranged Supp Fig 3 so that the panels a, b and c are now discussed in order, and edited the text as needed.

Reviewer #3 (Remarks to the Author):

The additional data provided by the authors did increase my concern that the authors are looking at an artifact that is caused by using qPCR as a tool to analyze telomere length.

Here are the main issues:

-The new data provided in figure 1J does not reflect the established telomere length seen in mice.

- an average telomere length loss of >70kb dramatically exceeds telomere length changes that are compatible with the telomere reserve present at the majority of mouse chromosome ends and inconsistent with the phenotype of the TERT KO mouse.

Response: We agree that the measurements obtained are unexpected but would like to point out that there are no previously published data on direct measurement of telomere length in the ICM of mice. The ICM telomeres are understood to be the longest of any somatic cells during the lifetime, and mouse telomeres are known to be particularly long. We have extensively validated our in-house qPCR assay and this is shown in the manuscript (Supp Fig 1). The second qPCR assay is a commercially available standardized test, and we have scrutinized the values that are generated and can find no reason to discount any of the datapoints. Importantly, the relative changes in telomere length are highly consistent between the two qPCR assays. Because of the lack of previous data in this area, we feel it is important to include these observations.

As a compromise to allay the Reviewer's concerns about the accuracy of this commercially available telomere assay kit, we have now moved this data to the Supplemental Figures, and edited the relevant text accordingly. We have also now included a new paragraph in the Discussion detailing the knowledge gap in this area and reasons for caution in interpreting the absolute telomere lengths.

We also conducted two additional experiments, in response to the Reviewer's comment.

- 1) We determined the amount of telomere lengthening that occurs between fertilization and ICM formation to provide the first baseline measurement of the extent of telomere elongation that occurs during embryo reprogramming. This experiment found that there is an extension of approximately 53kb during the phase of development between fertilization and ICM in the CBA.F1 strain of mice. Our colleagues at NIEHS (Dr Carmen Williams lab) have conducted an identical assay and found elongation of ~35kb between fertilization and blastocyst formation in the CF1 strain (data not shown). Our new data is now included as Supp Fig 2b and is highlighted in the text as supporting evidence of identical changes observed in our in-house qPCR assay and to provide foundational information on baseline measurements of telomere length in ICM.
- 2) We conducted an additional (fourth) assay of telomere length using the fetal DNA samples which are less limiting than the embryonic ICM. Telomere Restriction Fragment (TRF) assays were performed on a subset of the fetal heart DNA samples. The results on these few samples show the same trend as the two different PCR assays: reduced telomere length in the fetal hearts from rotenone-fed mice compared to controls. This data is now included as Supp Fig 7h.

New data: Supplementary Fig 2b and Supplementary Fig 7h.

New text (Discussion line 386-399):

Telomere length was measured by qPCR because it is the only technique currently feasible for measuring small amounts of DNA such as in single oocytes and embryos. We developed our own qPCR assay for relative telomere length, which was extensively validated (Supplementary Fig. 1, Supplementary Fig. 2) and avoids PCR artefacts from non-specific primer binding and pre-PCR amplification steps. To complement this approach, we also used a commercially available qPCR-based absolute telomere assay, to gauge the magnitude of changes in average telomere length; because previous studies of mouse ICM have only measured telomere length extrapolated from mean fluorescence intensity (Liu NCB 2007; Schaezlein PNAS 2004; Varela PNAS 2011). This method determined that chromosomes lengthen by approximately 50kb between fertilization and ICM formation (Supplementary Fig. 2b) which is not inconceivable since mice, especially inbred mice, are known to have particularly long and hyper-variable telomeres of up to 150kb in length in adult somatic cells (Kipling Nature 1990), and it is estimated that telomeres of mouse embryos lengthened by ~10kb in just the first cell cycle (Liu NCB 2007). Most importantly, both qPCR assays identified comparable relative changes between experimental groups in every model of mitochondrial manipulation.

- The authors use inbred mouse strains. It is well established that the telomere length differences within littermates of such inbred strains is much smaller than the telomere length differences reported by the authors. The large variations seen in the measurements within each experimental group thus provides a good estimate of the experimental accuracy of the qPCR assay.

Response: To clarify, our experiments primarily use CBA.F1 mice which are an outcross of two inbred strains, generated by crossing CBA females with C57 males (i.e. CBA/CaH x C57/BL6Arc). Eggs and sperm from these CBA.F1 mice were used to

generate the embryos used in Figure 1, and Figures 3-6. Experiments examining telomere length in obese and aged mice used the C57BL/6JSfdAnu-Alms1bbb/Abp strain of mice which is also not considered inbred. Mice from this colony were used to generate the embryos shown in Figure 2 and Figures 7-9. The only instances where inbred strains were used to create pure inbred embryos is shown in Supplementary Figure 2 where we conducted a comparison of telomere elongation in 5 different strains of mice to comprehensively validate our observations of telomere elongation kinetics during pre-implantation development. Similar results were observed using CBA.F1 mice (Fig 1b), CBAxC57, CBAxCBA, C57xC57, and C57SfdAnu (Supp Fig. S2 c-f) demonstrating that the greatest telomere elongation consistently occurs between the 8-cell and blastocyst stages.

Without an additional assay to measure telomere length the work is inappropriate for publication.

Response: Telomere length measurement is limited by the very small amounts of DNA available from individual embryos. In our manuscript, we have now employed a total of four different telomere length measurement assays: in-house qPCR, Q-FISH, Telomere Restriction Fragment (TRF) analysis, and a commercial qPCR kit. Each is useful for different indications, but in every case comparable results were observed in terms of the effects of mitochondrial stress on telomere elongation. We have highlighted these points in the new paragraph in the Discussion (lines 386-399).

Response to Reviewer comments

Reviewer 1 and Reviewer 2 were completely satisfied with the revised manuscript. Reviewer 3 and arbitrating Reviewer 4 queried the reliance on PCR-based assays and requested verification using another methodology. In response to these comments, we removed the data using the commercial qPCR assay in question and instead we conducted new experiments using a different methodology- telomere fluorescence in situ hybridization (telomere FISH).

The results of these experiments reproduce and confirm our validated qPCR assays. This new data, detailed below, is included in the revised manuscript and highlighted in the Tracked Changes document.

Point-by-point Response to Reviewer comments, reproduced verbatim:

REVIEWER COMMENTS

Reviewer #3 (Remarks to the Author):

The revision does not address my concern. The authors use a method to measure telomere length that has been shown by multiple studies to be highly inaccurate and to generate significant artifacts. While I appreciate that the authors attempt to validate their methodology, almost all key experiments rely on this PCR based method to measure telomere length.

Response:

We have now reproduced two of the key experiments using telomere FISH methodology instead of qPCR, as requested. New cohorts of embryos were generated from two different models of mitochondrial stress: exposure to 20% O₂ and exposure to rotenone. Telomere lengths in ICM cells were measured using telomere FISH.

This data is now included in the revised manuscript and shown as **Figure 1j** and **Figure 4k**. These results using telomere FISH reproduce and confirm the experiments that utilized qPCR.

1. **New Figure 1j** shows that embryos cultured in 20% O₂ have shorter ICM telomere length than embryos cultured in 5% O₂; confirming our previous result using qPCR (Fig. 1i).
2. **New Figure 4k** shows that blastocysts from mice fed rotenone diet have shorter ICM telomere lengths than blastocysts from mice fed standard diet; confirming our previous result using qPCR (Fig. 4j).

Reviewer #4 (Remarks to the Author):

Response to Reviewer comments

In this manuscript, the authors describe a significant effect of mitochondrial activity on telomere elongation between the 8-cell and blastocyst stages. Based on this data, the authors suggest that it is possible to modulate the preimplantation telomere resetting process using mitochondrially-targeted therapeutics (BGP-15, MitoQ, SS-31, metformin).

The principal concern with this study pertains to the use of qPCR to measure changes in telomere length. The authors do propose an alternative method for measuring telomere length, but all the key findings rely solely on qPCR. Coupled with the low amount of DNA used in most of these studies, the risk of artefacts is high. Whilst I appreciate that alternative methods for measuring telomere length are challenging in this context, I remain sceptical about the data presented in this study. The levels of telomere elongation observed are not compatible with the phenotypes described in telomerase-deficient mice.

A crucial experiment that would lend confidence to this study would be the analysis of telomerase-deficient mice. Examining changes in telomere length in the presence or absence of mitochondrial modulators would bolster confidence in the data provided. Without additional controls, I remain exceedingly sceptical of the changes in telomere length reported in this manuscript.

Response:

We have removed the data that utilized a commercial qPCR assay to determine absolute telomere length because the results were not reproducible. All of the qPCR data is generated via our in-house relative qPCR assay which is highly reproducible and which we confirmed does not produce artefacts and was validated by telomere FISH (Supplemental Fig S2).

We have now reproduced another two of the key experiments using telomere FISH methodology instead of qPCR. This data is now included in the revised manuscript and shown as **Figure 1j** and **Figure 4k**. These results using telomere FISH reproduce and confirm the experiments that utilized qPCR and demonstrate, using two different models, that mitochondrial activity influences telomere elongation during preimplantation embryogenesis. We have not repeated these experiments using telomerase-deficient mice since previous reports have already described the essential role of telomerase in embryo telomere elongation, and elucidating how mitochondrial dysfunction impinges upon the telomerase-mediated mechanisms is the subject of ongoing studies that are proving complex and beyond the scope of the current manuscript.

Response to Reviewer Comments

Reviewer 1 and Reviewer 2 were completely satisfied with the revised manuscript (version 2, January 2024). Reviewer 3 and arbitrating Reviewer 4 queried the reliance on a PCR-based assay and requested verification using another methodology (June 2024). While methods for telomere length measurement using low input DNA levels (e.g. a single zygote) are challenging, this was completed as requested. We conducted new experiments using telomere fluorescence in situ hybridization (telomere FISH), a non-PCR-based methodology. The results exactly reproduced two of our key findings, further validating our qPCR assay and substantiating our conclusions. This new data was included in a revised manuscript (November 2024) and is shown as Figure 1j and Figure 4k. Reviewer #4 has now requested that standards for the telo-FISH data be included. These are shown in the revised manuscript as a **new Supplemental Figure S2f**. We address in detail the remaining Reviewer comments below.

REVIEWERS' COMMENTS

Reviewer #3 (Remarks to the Author):

My fundamental concern with this work remains unresolved. The authors fail to convincingly establish that the measured telomere length changes in this paper are physiologically meaningful. This issue arises primarily because the work relies on a telomere length measurement tool that has been demonstrated to produce artifacts. Despite several rounds of revision, this criticism has not been adequately addressed. The piecemeal approach of iteratively making incremental changes to the telomere length measurements has not yielded data that support the broad claims made in this paper.

Response: The Reviewer is wrong to suggest that we are using a tool 'that has been demonstrated to produce artifacts.' To be clear, we established our method with awareness of some flaws in similar published methods and thus extensively validated our qPCR assay, to convincingly show it does not generate artifacts (Supp Fig S1 and detailed below). Next we performed calibration tests to show our assay faithfully recapitulates verified telomere length dynamics (Figs 1b, Fig 1c, Supp Fig S2b-e) that are already well-documented by others. Thirdly, we conducted independent experiments using a different, non-PCR-based, methodology: telomere fluorescence in situ hybridization (telo-FISH). The results of these telo-FISH assays (Fig 1j, Fig 4k, Supp Fig 2a,) exactly reproduced the qPCR measurements.

We are well aware, as the Reviewer also appears to be, that some telomere qPCR assays described in the published literature do indeed create artifacts. We discovered this from our own experience and have not wanted to fixate on these issues previously, but now feel compelled to further describe these findings in order to defend the integrity of our own qPCR assay.

Response to Reviewer Comments

- 1) We originally trialled a widely used assay for measuring telomere length in mammalian embryos (Wang et.al 'Robust measurement of telomere length in single cells' PNAS (2013) <https://doi.org/10.1073/pnas.1306639110>). This assay involves a preamplification step with both the telomere and m18S rRNA (reference gene) primers followed by product purification and a subsequent qPCR using either telomere or m18S rRNA primers. We found that this assay produced significant artifact products during the pre-amplification step. Specifically, PCR products were generated even in the negative controls (H₂O instead of sample). As such, **no pre-amplification step is used in our method. Negative controls are always included in our assays and artifacts are not observed.**
- 2) We found that the reference gene primers used in other assays are not suitable and contribute to artifacts. Specifically, when we scrutinized the m18S rRNA primers used by Wang et.al PNAS (2013) using NCBI primer BLAST, we found that multiple products of different sizes were generated and only a small fraction of these were an identity match to the m18S rRNA gene. We next tested Rn18S primers published by de Frutos C, et al. (1). **These Rn18S primers return a consistent product size (91bp), and alignment to the mouse genome found 90% of alignments had 100% query coverage and identity match. These validated Rn18S primers are used to detect the reference gene in our assay.**
- 3) We also trialled a commercially available qPCR assay kit (ScienCell Absolute Mouse Telomere Length Quantification qPCR Assay Kit, Catalog #M8918) to measure telomere length. This assay includes reference DNA of 'known telomere length' to enable calculation of absolute telomere length. Unfortunately, we found inconsistencies in the measurements produced by different kits. Specifically, analysis of the exact same sample (DNA purified from mouse skin biopsy) using four different kits returned absolute telomere length measurements that varied by >130kb. We further determined that the inconsistencies were due to differences in the Ct values of the reference gene standard that is critical to calibrate the amount of input DNA/nuclei. **Because of the non-reliability to measure absolute telomere length using commercial kits, we have only measured relative telomere length in the current manuscript. Other key studies in this area also measure relative telomere length (2-5).** Future work to determine absolute telomere lengths, is well beyond the scope of this manuscript. The physiological significance of relative telomere length differences is widely studied and is a foundational principle, but not the subject of this study.

Thus, even though the Reviewer has only provided vague aspersions of our work, without recognizing the careful benchmarking we performed, nor pointing to any sign of artifact in the data, or specifying a more preferred method, we acknowledge the underlying point and have made every effort, repeatedly, to provide validation that addresses any concerns about our qPCR assay. A detailed description of our

Response to Reviewer Comments

methodology is to be published as a peer-reviewed chapter in “The Pre-Implantation Embryo” edition of the lab protocol series **Methods in Molecular Biology**, to be published in 2025 by Springer Nature. We have already shared this protocol with our colleagues who have used it to measure telomere length in embryos and replicated our results. We would encourage others to employ this assay and determine for themselves that it does not generate artifacts and is highly reproducible.

Reviewer #4 (Remarks to the Author):

The revised manuscript fails to address my previous. In my previous revision, I suggested additional experiments to address the issues associated with qPCR as a method to measure telomere length. Specifically, I suggested:

To use an alternative method for telomere length measurement:

While the authors have included FISH quantification to support the qPCR data, this addition remains insufficient. FISH quantification without appropriate controls—such as cell lines with known telomere lengths as standards—makes the interpretation of the data extremely challenging. Without these controls, the reliability of the FISH measurements cannot be adequately assessed.

Inclusion of telomerase KO mice as controls: I suggested to include telomerase KO mice as a control aimed validating the reported changes in telomere length. The authors decided not to perform these experiments since “previous reports have already described the essential role of telomerase in embryo telomere elongation”. This is reasoning is perplexing: this is precisely the reason, I suggested including this control.

Response: We have now included controls used in the telo-FISH assay, as requested. This data is shown as **new Supp Fig S2f**. These controls are cell lines known to have homogeneous short telomere lengths (HCT116, HeLA; average 4-5kb) or long heterogeneous telomeres (U-2 OS; average ~36kb), and which are used to demonstrate the fidelity of the telo-FISH assay. As expected, and as shown in **Supp Fig S2f**, telomere length (fluorescence per cell) is lowest in HCT116 cells and in HeLa cells and an order of magnitude higher in U-2-OS cells. These controls confirm the reliability of the FISH measurements. This methodology is a widely accepted standard, published by our team (6) and others in the field (7).

We did not pursue the Reviewer’s suggestion to measure telomere length in telomerase KO mice. This is because the result would be only one more replication of the standards described above and because it has been previously published that telomeres are shorter in telomerase KO mice (2). We felt it did not justify the inordinate amount of time and welfare issues required to import mice and establish a colony simply to repeat this result. Instead, there are other aspects of our data

Response to Reviewer Comments

that provide equally valuable validation of our assay as the singular experiment suggested by the Reviewer. 1) The lengthening of telomeres during mouse preimplantation embryogenesis that has been reported by others using a combination of qPCR and FISH (2,3) was also observed by us (Figure 1b, Supplementary Fig. S2 a-e). 2) Longer telomere length in ICM versus TE that has been reported by others (4,5) was also demonstrated in our work (Fig 1c). These results validate that our qPCR assay provides identical results to those published by others.

Our methods description is updated to now include the assay standards. **New text** (lines 680-682): 'Standards for this assay (see Supplementary Fig. S2f) are cell lines known to have homogeneous short telomere lengths (HCT116, HeLA) or long heterogeneous telomeres (U-2 OS).'

A relevant paragraph in the Discussion has also been updated (new text is underlined below) to better highlight the consistency between our qPCR assay and the FISH assay.

Lines 383-385: We further used quantitative telomere FISH which we and others (12-14) have previously established to measure mouse ICM telomere lengths, and replicated key findings in independent cohorts of embryos using this second methodology. Importantly, both assays verified the expected telomere elongation during preimplantation embryogenesis and also identified comparable relative changes between experimental models of mitochondrial stress and stress reversing treatments.

REFERENCES:

1. de Frutos C, et al. (2016) 'Spermatozoa telomeres determine telomere length in early embryos and offspring' *Reproduction*. 151(1):1-7. doi: 10.1530/REP-15-0375.
2. Liu, L., Bailey, S.M., Okuka, M., Muñoz, P., Li, C., Zhou, L., Wu, C., Czerwiec, E., Sandler, L., Seyfang, A., Blasco, M.A., Keefe, D.L., 2007. Telomere lengthening early in development. *Nature Cell Biology* 9, 1436–1441. <https://doi.org/10.1038/ncb1664>
3. Schaetzlein, S., Lucas-Hahn, A., Lemme, E., Kues, W.A., Dorsch, M., Manns, M.P., Niemann, H., Rudolph, K.L., 2004. Telomere length is reset during early mammalian embryogenesis. *Proceedings of the National Academy of Sciences* 101, 8034–8038. <https://doi.org/10.1073/pnas.0402400101>
4. Iqbal, K., Kues, W.A., Baulain, U., Garrels, W., Herrmann, D., Niemann, H., 2011. Species-Specific Telomere Length Differences Between Blastocyst Cell Compartments and Ectopic Telomere Extension in Early Bovine Embryos by Human Telomerase Reverse Transcriptase. *Biology of Reproduction* 84, 723–733. <https://doi.org/10.1095/biolreprod.110.087205>

Response to Reviewer Comments

5. Varela, E., Schneider, R.P., Ortega, S., Blasco, M.A., 2011. Different telomere-length dynamics at the inner cell mass versus established embryonic stem (ES) cells. *Proceedings of the National Academy of Sciences* 108, 15207–15212. <https://doi.org/10.1073/pnas.1105414108>
6. Lu R, Nelson CB, Rogers S, Cesare AJ, Sobinoff AP, Pickett HA (2024) *iScience* 27(1):108655 'Distinct modes of telomere synthesis and extension contribute to Alternative Lengthening of Telomeres'
7. Publications (in *Nature Communications*) that use telomere FISH to measure telomere length:
<https://doi.org/10.1038/s41467-019-11785-7> (Fig 3);
<https://doi.org/10.1038/s41467-019-12664-x> (Fig 5, Fig 6)
<https://doi.org/10.1038/s41467-019-08863-1> (Fig 6)